# The use of spatial data and satellite information in legal compliance and planning in forest management

**Chris Taylor, David B. Lindenmayer** [ORCID]*

Fenner School of Environment and Society, The Australian National University, Canberra, ACT, Australia

* david.lindenmayer@anu.edu.au

**Data Availability Statement:** The data underlying the results presented in the study will be available from (TBC).

**Funding:** The authors received no specific funding for this work.

## Abstract

A key part of native forest management in designated wood production areas is identifying locations which must be exempt from logging. Forest laws, government regulations, and codes of practice specify where logging is and is not permitted. Assessing compliance with these regulations is critical but can be expensive and time consuming, especially if it entails field measurements. In some cases, spatial data products may help reduce the costs and increase the transparency of assessing compliance. However, different spatial products can vary in their accuracy and resolution, leading to uncertainty in forest management. We present the results of a detailed case study investigating the compliance of logging operations with laws preventing cutting on slopes exceeding 30˚. We focused on two designated water catchments in the Australian State of Victoria which supply water to the city of Melbourne. We compared slopes that had been logged on steep terrain using spatial data based on a Digital Elevation Model (DEM) derived from LiDAR, a 1 arc second DEM derived from the Shuttle Radar Topography Mission, and a Digital Terrain Model (DTM) with a resolution of 10m. While our analyses revealed differences in slope measurements among the different spatial products, all three datasets (and the on-site slope measurements) estimated the occurrence of widespread logging of forests on slopes >30˚ in both water catchments. We found the lowest resolution Shuttle Radar Topography Mission DEM underestimated the steepness of slopes, whilst the DTM was variable in its estimates. As expected, the LiDAR generated slope calculations provided the best fit with on-site measurements. Our study demonstrates the value of spatial data products in assessing compliance with logging laws and codes of practice. We suggest that LiDAR DEMs, and DTMs also can be useful in pro-active forest planning and management by helping better identify which areas should be exempt from cutting before logging operations commence.

## Introduction

Natural forests worldwide have numerous cultural, social, economic and environmental values. These values can include significance to Indigenous Peoples, provision of water for

**Competing interests:** The authors have declared that no competing interests exist.

human consumption, carbon storage, extraction of timber, and supporting habitat for vast numbers of species [1]. Managing these sometimes competing values can be complex, demanding sophisticated forest management planning and compliance with laws and codes of management practice [2,3]. These laws, codes of practice and other relevant regulations specify, for example, the volume of timber that can be extracted from a forest and the locations where logging is permitted to occur [4–10]. For example, many nations have codes of practice regulating the width of streamside reserves or buffer strips to protect water catchment condition in wood production forests [4,6,11–13].

A key part of forest management is identifying where logging is and is not appropriate [6]. Logging exemption areas can include riparian zones, areas of high environmental risk such as those on steep terrain or where there is an increased risk of soil erosion, areas of high biodiversity and conservation value, and sites of cultural significance [8]. Ensuring that agencies responsible for logging operations are compliant with regulations is an essential part of ecologically sustainable forest management. However, determining whether there is compliance with regulations can be expensive and time consuming, especially if it entails measurements in the field [14]. In some cases, the use of various spatial data products may help reduce the costs and increase the transparency of assessing compliance with codes of practice for environmental management [15].

There is a long history of forest management planning, particularly as specified under codes of forest practice [2,16]. A key part of such planning is monitoring compliance with codes of practice and associated forest law [17]. Some of the most cost-effective methods of monitoring compliance across large forest areas include the use of satellite and other remote sensing data [18] as well as spatial data modelled from a satellite–based or otherwise air-borne active sensor giving rise to digital elevation models (DEMs). DEMs can assist in identifying steep slopes and other topographical features [19,20]. However, DEMs can vary in their accuracy and resolution leading to differences in, for example, the calculations of slope derived from them [19,21]. Such differences may make it difficult to assess levels of compliance of logging operations with laws and prescriptions in forest management areas [22]. Indeed, some regulatory authorities have argued that DEMs overestimate the steepness of slopes [23] and therefore logged areas thought to be in breach of forestry laws and prescriptions using one method may not be under other approaches.

In this study, we quantified slope calculations derived from three DEMs in logged areas within two water supply catchments located in the forests of Victoria (south-eastern Australia). Codes of forest practice prohibit the logging of slopes over specified limits [8,24] within designated water supply catchments. The use of DEMs is essential for forest management planning to identify steep slopes and exclude logging from these areas. However, there are multiple DEM products at different scales and that cover different areas. We compared a DEM generated from LiDAR, a 1 arc second (DEM) derived from the Shuttle Radar Topography Mission, and the VicMap Elevation Digital Terrain Model (DTM) 10m, which was generated by the Victorian Government [25]. LiDAR is often recognized as having very high spatial resolution and accuracy [19,26]. However, many areas of forest have not yet been surveyed using high resolution LiDAR; lower resolution options, such as the Shuttle Radar Topography Mission (SRTM) derived DEM, are used in these areas [27]. Furthermore, high resolution LiDAR data are not always open access and obtaining these data can may be difficult for some stakeholders [20]. It is therefore important to assess how different DEMs compare and to assess the accuracy of slope derived calculations.

In this study, we asked the following two questions: **(1)** *What are the levels of congruence among slope calculations across cut blocks derived from different Digital Elevation Models*? And, **(2)** *How do these calculations compare with on-site slope measurements*? Our data collection

was limited to logged areas in water supply catchments servicing the city of Melbourne and surrounding rural areas. We compared the extent of the area calculated for slopes greater than 30˚ identified by the three DEM models and on-the-ground measurements. At the time of our analysis, government laws in the State of Victoria stipulated that slopes above 30˚ must be excluded from logging in water supply catchments [24]. Ensuring compliance with regulation is important to protect water supply catchments from erosion and to maintain the critical role of forests in the provision of water for human consumption [28]. Our investigation provides insights into the accuracy of slope assessments in forests where the only available spatial data are relatively low resolution DEMs. We discuss how spatial data products can be used to make a significant contribution to both forest management planning and compliance with codes of practice that are allied to such planning.

## Methods

### Study area

We focused our analysis on 15 forest management blocks which form part of two water supply catchments that provide water to the city of Melbourne and surrounding rural regions. These areas were the Upper Goulburn and Thomson water supply catchments (Fig 1). Under Victorian Government forestry regulations, the Upper Goulburn River Catchment has been declared a *Water Supply Protection Area* [24]. This catchment supplies water to the Eildon Reservoir, which then provides water to rural regions in Victoria and to the city of Melbourne via the north-south pipeline [29]. The catchment is 279,143 ha in size [30], of which 85% is forest. The land tenures in the Upper Goulburn River Catchment are State Forest (67%), private land/infrastructure/other tenure (22%), and conservation reserve (11%). The Thomson River Catchment supplies water to the Thomson Reservoir, the largest reservoir in the Melbourne Water Supply system [31]. This reservoir has a capacity of 1,068 billion litres, comprising 59% of Melbourne's total water storage capacity [32]. The Thomson River catchment is a designated *water supply protection area* under Victorian Government forestry regulations [24]. The catchment is 48,371 ha in size [33], of which 95% is forest. The land tenure of this catchment consists of state forests (85%), private land/infrastructure/other tenure (8%), and conservation reserve (7%).

We analysed the 15 forest management blocks for which there was Digital Elevation Model (DEM) data coverage. These consisted of 10 forest management blocks in the Upper Goulburn water supply protection area and six forest management blocks in the Thomson water supply protection area, with one forest block shared between the two catchments. These forest management blocks covered an area of 77,668 ha and 41,591 ha in the Upper Goulburn and Thomson water supply protection areas, respectively.

### Forest management and codes of forest practice

We focused our analysis on clearfell logging that occurred between 2004 and 2020 in the Upper Goulburn and Thomson Catchments. This 16-year period captured all logging conducted by VicForests (which was declared a Victorian Government business corporation in late 2003) in the Upper Goulburn and Thomson Catchments [34].

Logging in State Forest is required to comply with the *Code of Forest Practices for Timber Production* [8] and its incorporated document, the *Management Standards and Procedures* [24] under the provisions of the Sustainable Forests (Timber) Act 2004. At the time of our analysis, the 2014 Management Standards and Procedures was applicable [24]. Under Section 3.5 of this version of the *Management Standards and Procedures*, logging on steep slopes was addressed and it stated that logging was prohibited in all areas of forest occurring slopes >30˚

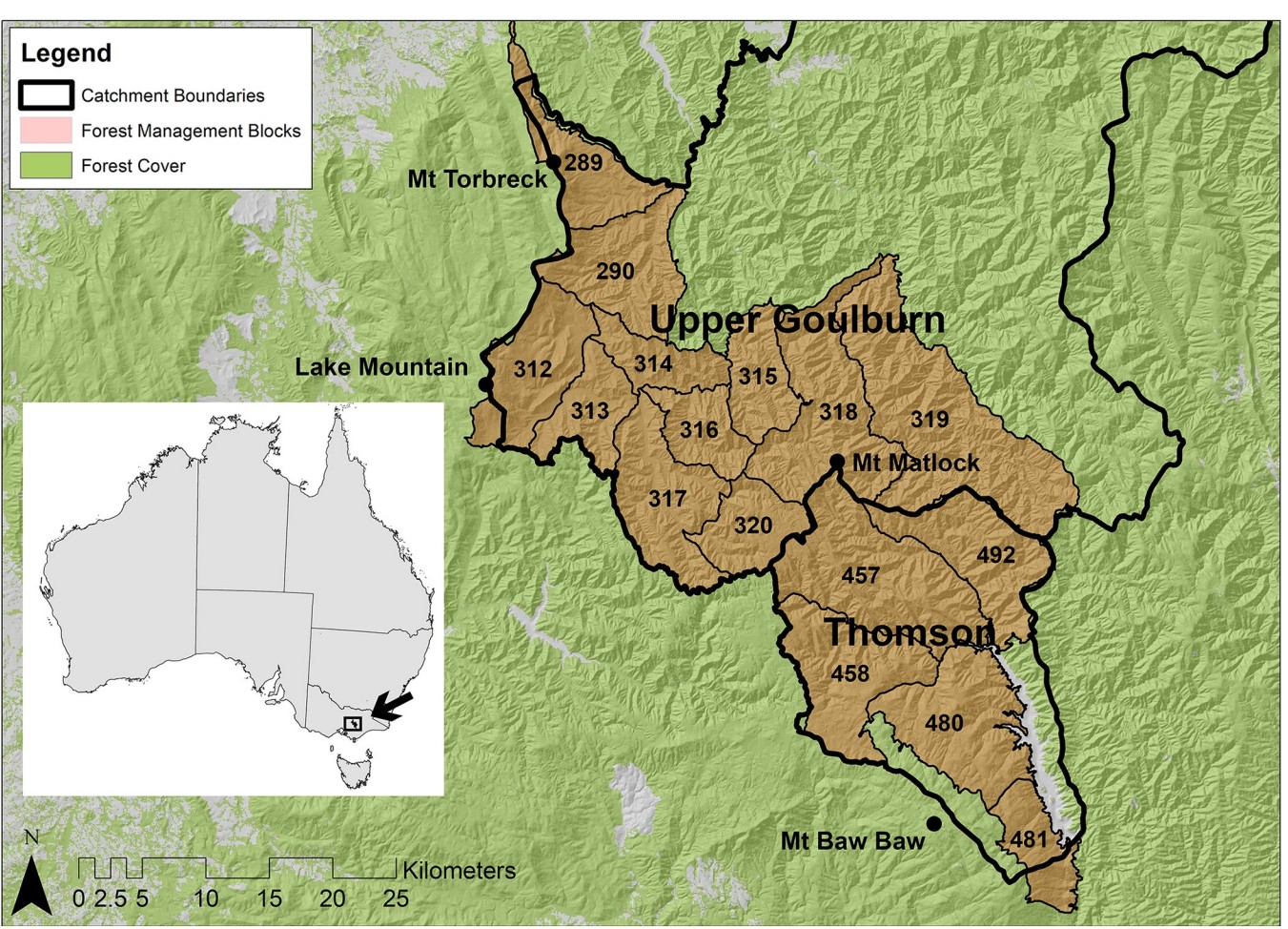

**Fig 1. Location of study area consisting of forest management blocks (identified by number) across the Upper Goulburn and Thomson water supply protection areas.**

across listed water supply protection areas [24]. These areas included the Upper Goulburn and Thomson water supply protection areas [24]. However, these Management Standards and Procedures did not prescribe a minimum patch size [24]. To address ambiguities surrounding compliance, the Office of the Conservation Regulator (OCR) provided an explanation for a minimum area of slope >30˚ being *the average slope of the topographic feature of interest that is subject to assessment* [23,35]. However, this explanation did not specify a minimum distance. Under a subsequent revision of the Management Standards and Procedures following our analysis, the Victorian Government weakened this prescription to allow for up to 10% of logged areas to occur on slopes >30˚ inside water supply protection areas [36].

## Slope analysis using digital elevation models, digital terrain models and LiDAR

To conduct comparative slope analyses, we used a DEM generated by the Shuttle Radar Topography Mission (SRTM) [27], the VicMap Elevation Digital Terrain Model (DTM) with a resolution of 10m [37], and LiDAR data sourced from the Central Highlands LiDAR survey project [38] which was obtained through a Freedom of Information Request by the authors to VicForests [39]. We obtained the DEM generated by the Shuttle Radar Topography Mission

from Geosciences Australia [40]. This dataset has a resolution of 1 arc second (~30m) and consisted of a bare-earth DEM with a regular grid representing ground surface topography. It excluded, where possible, features such as vegetation and human-made structures. The SRTM data is part of a global dataset released by US Defence Department in the WGS84 projection. For Australia, the horizontal positional error for the SRTM DEM is within 7.2 m and the elevation error is within 9.8 m [40].

The VicMap Elevation Digital Terrain Model (DTM) 10m was developed as a joint project through the National Action Plan for Salinity and Water in conjunction with the then Department of Sustainability and Environment (DSE) and Sinclair Knight Merz [25]. The methodology employed to develop this dataset used both ANUDEM [41] and Triangulated Irregular Network (TIN) [25] processes. It included inputs from VicMap Elevation, VicMap Hydro, LiDAR and Photogrammetry derived DTMs and contours, Survey Marks Enquiry Service, and the DEM from the Shuttle Topography Radar Mission [25]. This dataset has a resolution of 10m and the raster pixels in this dataset represent the average elevation of the pixel. This dataset has been used to calculate slopes >30˚ by VicForests across its operations mapping [42–44]. The datum used in the construction and maintenance of VicMap Elevation DTM 10m is the Geocentric Datum of Australia. The dataset has a positional accuracy of ±12.5m horizontally and ±5m vertically or better (GDA94) [25].

We sourced the LiDAR data from the Central Highlands LiDAR survey project [38]. This project used elevation data from a 4,580 km$^2$ area northeast of Melbourne in Victoria. The elevation data were captured between January and May 2016 [37] and covered all the Thomson water supply protection area and part of the Upper Goulburn water supply protection area. LiDAR data are generated by airborne LiDAR sensors taking discrete return measurements [26]. LIDAR uses electromagnetic waves in the optical and infrared wavelengths through an active sensor, where electromagnetic waves are emitted, and the sensor receives the reflected signal back [45]. Data are captured using an automatic classification algorithm, where LiDAR points were classified under categories such as ground and first return [38]. In the Central Highlands LiDAR survey project, TerraScan software was used by DELWP [38] to produce an initial classification of the terrain, where ground points were confirmed and retained. The data were stored in LASer or LAS format [38], which is a file format for the interchange of 3-dimensional point cloud data [46], and arranged in a grid of 1km-by-1km LAS files. The dataset has a positional accuracy of ±0.2m horizontally and vertically [38].

We generated slope rasters using the slope tool in ArcGIS [47,48] from the SRTM DEM [27], VicMap Elevation DTM 10m [38], and the DEM we generated from LiDAR [38]. This tool accounts for slope gradients in both the x and y directions. The slope raster derived from the SRTM DEM was generated with the World Geodetic System (WGS) 1984 [37] as the coordinate system, which required conversion from metres to arc seconds for its Z factor. We set our Z factor parameter to 37˚30'S, which is the approximate latitude of our study area, to generate our slope raster in degrees from the SRTM DEM. The slope raster derived from VicMap Elevation DTM 10m was generated in degrees using the Geocentric Datum of Australia 1994 Adjustment (GDA94), the native projection of the dataset [25]. We processed the LiDAR LAS files into floating DEM rasters at a resolution of 1m in ArcGIS by identifying ground points and binning them using an average cell assignment type [26]. The point separation ranged between 10 cm to 25 cm. We determined the value of a raster cell by examining the LiDAR points that were located within that cell. We calculated the average value of multiple points within each raster cell. Where cells did not have points, we triangulated across void areas and used linear interpolation on the triangulated value to determine the cell value [49]. We generated a DEM consisting of a floating raster across the Upper Goulburn and Thomson water supply protection areas in ArcGIS [47] at 1m resolution. We generated this slope raster in

degrees using the MGA 1994 Zone 55 projection, which was the native projection of the LAS files [30]. As LiDAR and VicMap Elevation DTM used projected coordinate systems, the Z factor was set a 1.0.

Data generation and collation provided us with three primary datasets: the STRM DEM Slope Raster at 1 arc second (~30m) resolution (SRTM Slope), the VicMap Elevation DTM Slope Raster (VicMap Slope) at 10m resolution, and the LiDAR generated slope at 1m resolution (LiDAR 1m Slope). The LiDAR generated slope raster was inclusive of very small areas of slopes >30˚, including topographical features as small as $1m^2$. As these areas would not necessarily constitute a breach of the prescriptions under Clause 3.5 of the 2014 Management Standards and Procedures, we applied an average slope calculation across the cutblock areas, as guided by the descriptions of *average slope* provided by the OCR [23,49].

To estimate the area of *average slope* >30˚ across a *topographic feature of interest* [23,49], we generated a second slope raster generated from the LiDAR DEM using average slope neighbourhood areas [50,51]. This approach was based on using focal statistics to calculate the average slope value across a specified *neighbourhood* area around each raster cell throughout the water supply protection areas. The OCR did not specify an minimum distance describing an average distance to measure a steep slope [23,49]. In the absence of such a specified minimum distance, we used a moving circle *neighbourhood* average area consisting of a 5m radius, where each cell in the centre of a circle was the average slope value of the circular neighbourhood of cells surrounding it. By using this average neighbourhood slope method, we identified where there were average slope areas >30˚ for an average distance of at least 10m across cut blocks in the Upper Goulburn and Thomson water supply protection areas. We did not use average neighbourhood calculations for our slope calculations generated from the VicMap Elevation DTM and the SRTM DEM, because these were at cell sizes of 10m and ~30m, respectively. We contend that slope values >30˚ across these respective cell sizes would indicate an average slope.

In total, we generated four slope rasters for our analysis: **(1)** Slope Raster at 1m resolution (LiDAR 1m), **(2)** Average Neighbourhood Slope (LiDAR F5m), consisting of the average slope value within 5m radius of a cell, **(3)** Slope Raster VicMap Elevation DTM at 10m resolution (DTM). And **(4)** Slope Raster SRTM at ~30m resolution (SRTM). We grouped all slope rasters into slope values 0˚to 10˚, 10˚ to 18˚, 18˚ to 27˚, 27˚ to 30˚ and >30˚. These class intervals are based on the *Absolute Risk Rating Methodology* previously used in the Forest Audit Program for the Victorian Government [52].

## Forest management data input

We derived logging history information across the study area using the dataset LOG_SEASON [53]. This dataset contains the details of the time an area was known to be logged and the logging method employed. We identified cutblocks within this dataset for analysis. Cutblocks were defined as areas that were logged using the silvicultural methods of clearcutting, seed tree, regrowth retention and clearcutting salvage. These forms of logging involve the removal of all merchantable logs in a single integrated harvesting operation [54]. Remaining slash and forest debris is then partially consumed in a high-intensity planned burn following logging [55]. Cutblocks were identified by address, which consisted of the forest block number, compartment number and the cutblock number. The sourced logging history dataset contained multiple polygons for many of the cutblocks. We combined these to form one polygon per cutblock address.

We grouped cutblocks into the percentage of the respective net areas containing slopes >30˚. These consisted of: **(1)** All areas <30˚ in slope; **(2)** 0–1% of cutblock area >30˚ in slope;

**(3)** 1–5% of cutblock area >30˚ in slope; **(4)** 5–10% of cutblock area >30˚ in slope; and **(5)** >10% of cutblock area >30˚ in slope. This allowed us to differentiate cutblocks with relatively large areas of slope >30˚ from those cutblocks consisting of smaller areas with slopes exceeding >30˚. We identified the areas of Upper Goulburn and Thomson water supply protection areas by using the dataset *Designated Water Supply Catchments* [33]. We also used forest management blocks to analyse subset areas within the water supply protection areas [56].

## Field observations

In response to our past work which suggested that forests on steep slopes had been logged [57], the Timber Harvest Compliance Unit of the Office of the Conservation Regulator (OCR) investigated two cutblocks in the Upper Goulburn water supply protection area, where it measured a total of 38 transects, which ranged from 14m to 102m in length [22,23]. This provided us with an opportunity to compare a series of on-site measurements reported by the Timber Harvest Compliance Unit with our LiDAR-derived slope calculations [22,23]. We plotted 36 of the 38 transects onto the slope rasters we generated from the LiDAR DEM [38], VicMap Elevation DTM [25] and the SRTM DEM [27]. We calculated the average slope value of these transects. These transects were grouped as the OCR transects. We also measured the slopes of 26 transects across three additional cutblocks and plotted these onto our slope rasters. These transect lengths ranged from 10m to 34m. This involved the use of methods described in the VicForests' Utilisation Procedures [58]. We measured the transects using a Nikon Laser Rangefinder Forestry Pro II. We selected cutblocks where post-logging regrowth did not obstruct the line of site between each end of the transect. We grouped these transects (termed the "ANU transects") and measured slope angles in degrees. In total, we used 62 transects to compared on-site measured slopes with slope calculations generated from the LiDAR DEM, VicMap Elevation DTM and the SRTM DEM.

## Data analysis

We analysed the slope rasters generated from the SRTM DEM, VicMap Elevation DTM, and LiDAR DEM by comparing elevation and slope differences between the datasets. As each DEM and DTM were at different resolutions no greater than ~30m, we created a 100m grid of points. We extracted elevation and slope data from each DEM/DTM and each slope raster and analysed data for points located within cutblocks logged between 2004 and 2019. Using a grid of points at 100m avoided replication of data, particularly for the 1 arc second SRTM DEM where raster resolution was ~30m. We used the LiDAR DEM and generated slope raster as a baseline because it contained the highest accuracy of the datasets analysed (±0.2m) [30]. Following the methods used by DeWitt et al. (2015) [21] and Gonga-Saholiariliva et al. (2011) [59], we generated a datum for each site by subtracting the LiDAR DEM from each DEM and DTM (Each DEM-LiDAR DEM) and the LiDAR DEM generated slope from each DEM and DTM generated slope (Each Slope—LiDAR Slope). In the difference data, positive values indicated areas where the evaluated non LiDAR DEM (VicMap Elevation DTM or SRTM DEM) surface and slope was above that of the baseline LiDAR DEM and LiDAR Slope, respectively, while negative values indicate areas below that of the baseline LiDAR surface and slope [21].

We used the Shapiro–Wilk (W) test to evaluate the DEM value distributions, with the null hypothesis being a normal distribution. We also used QQ Plots to further assess whether our data aligned with a normal distribution. Where our data was aligned with a normal distribution or contained a large volume of data, we used the paired Student's T Test to compare the data. Where our data did not align with a normal distribution and it consisted of a small volume of data, we used the paired Wilcoxon Signed Rank Exact test. We used boxplots to

evaluate the variability of the differences between the datasets, and whether they were predominantly positive or negative [21]. Difference measures of central tendency for each DEM, DTM and slope raster, such as the mean and the median, provided insight into how each DEM, DTM and slope raster compared to the baseline LiDAR DEM and the two LiDAR generated slope rasters (whose mean and median difference values were assumed to be near-zero). We analysed standard deviation around the mean, where a high standard deviation indicated increased variation between the DEMs and generated slope rasters [60].

We compared each slope raster from the LiDAR, VicMap Elevation DTM, and SRTM DEM to on-site slope measurements, both from the authors (ANU Transects) and the Timber Harvest Compliance Unit of the OCR (OCR Transects) [22]. We used the paired Wilcoxon Signed Rank and the Student's T tests to compare the median of the observed measurements with the median of the slope calculations generated from the LiDAR DEM, VicMap Elevation DTM, and SRTM DEM. We also compared the area >30˚ for each cut block calculated from the slope rasters generated from the LiDAR DEM, VicMap Elevation DTM, and SRTM DEM using the paired Wilcoxon Signed Rank Exact Test.

## Results

We generated data for 4621 points arranged within a 100m grid in cutblock areas in the Upper Goulburn and Thomson water supply protection areas. Although our input data was not normally distributed for our analyses of elevation, slope and areas >30˚ in slope across cutblocks (S1 Table), it consisted of a large number of sites (4621), so we used the paired Student's T Test to test for significance. We found significant differences (P<0.05) in elevation between the LiDAR 1m DEM, the SRTM DEM, and the VicMap Elevation DTM (S2 Table). The SRTM DEM had higher elevation values compared to the LiDAR 1m DEM and the VicMap Elevation DTM was lower (Fig 2). The largest difference was between the LiDAR 1m DEM and the SRTM DEM, where we identified the mean of the SRTM DEM being 22.3m higher than the LiDAR 1m DEM (S3 Table). The elevation mean of the VicMap Elevation DTM was lower by 0.89m compared with the LiDAR 1m DEM. The standard deviation in difference (10.45m) between the LiDAR 1m DEM and the VicMap Elevation DTM was larger than its mean

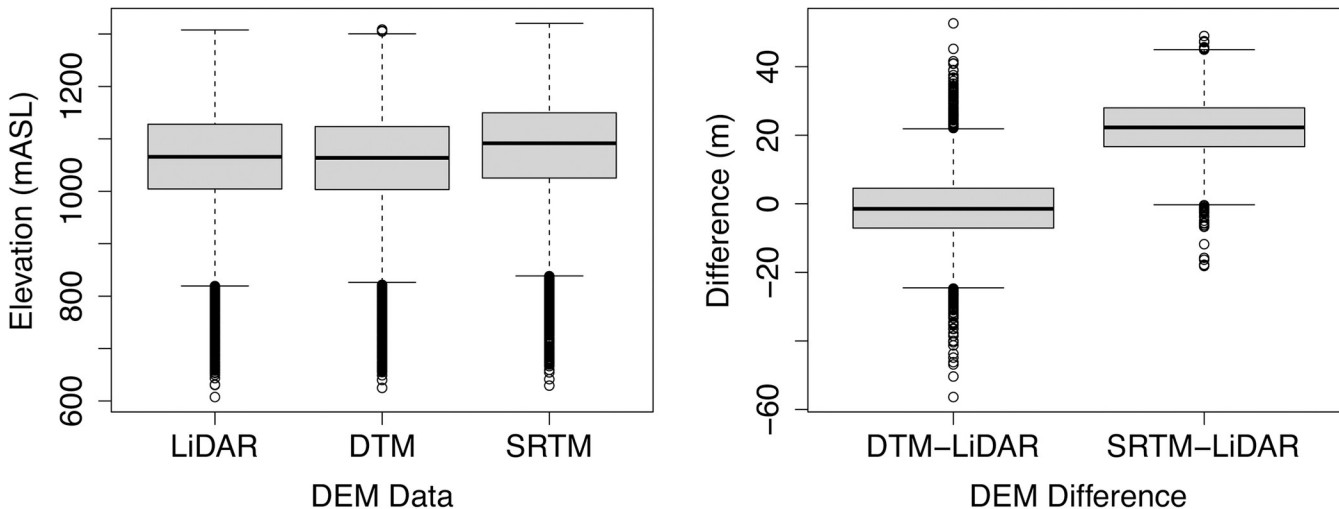

**Fig 2. Elevation range in metres above sea level (mASL) for the LiDAR 1m DEM, VicMap Elevation DTM and SRTM DEM (left) and the difference in elevation between LiDAR 1m DEM and the VicMap Elevation DTM (LiDAR 1m-DTM) and SRTM DEM (LiDAR 1m-SRTM).**

difference and the standard deviation was smaller than the mean difference between the LiDAR 1m DEM and SRTM DEM (8.64m).

We used a paired Student's T Test to compare the LiDAR generated slope rasters with the slope rasters generated from the VicMap Elevation DTM and the SRTM DEM. We found significant (P<0.05) differences between the slope rasters calculated from the LiDAR 1m DEM, VicMap Elevation DTM, and SRTM DEM (S4 Table). These consisted of the Slope raster LiDAR 1m, LiDAR F5m (average slope neighbourhood area with a 5m radii), and slope rasters generated from the VicMap Elevation DTM and SRTM DEM (Fig 3). The slope calculations generated from the VicMap Elevation DTM and the SRTM DEM were lower than those generated by the LiDAR across most sites. Mean differences in slope ranged between -0.8˚ to -1.01˚ between the datasets (S5 Table). However, there was variation across these differences, with standard deviations ranging from 5.91˚ to 8.25˚.

## Cutblock slope comparisons for the Upper Goulburn and Thomson water supply protection areas

The slope raster (LiDAR 1m) we generated from LiDAR DEM showing all slopes >30˚ indicated that all cutblocks in the Thomson water supply protection area contained slopes >30˚ and all but two cutblocks supported slopes >30˚ in the Upper Goulburn water supply protection area (Table 1). We calculated that 179.0 ha of forest logged in the Upper Goulburn water supply protection area was on slopes >30˚. The equivalent area for the Thomson water supply protection area was 102.4 ha. These values equate to 7.1% and 4.9% of the total area logged for the Upper Goulburn and Thomson water supply protection areas, respectively. The area of some of these slopes >30˚ area was small and may have included areas ~ 1m$^2$. However, 41 of 154 cut blocks in the Upper Goulburn water supply protection area and 23 of 144 cutblocks in the Thomson water supply protection area had >10% of their respective areas occurring on slopes >30˚. When we applied the average neighbourhood slope area of 5 metres radii to our slope raster generated from the LiDAR DEM (LiDAR F5m), the areas of slope calculated >30˚ decreased to 78.9 ha and 37.4 ha respectively for the Upper Goulburn and Thomson water supply protection areas. We calculated that 11 cutblocks in the Upper Goulburn and five in the Thomson water supply protection areas had >10% of their respective areas occurring on slopes >30˚. For our slope raster generated from the SRTM 1 arc sec DEM (SRTM), we calculated that 91 and 63 cutblocks in the Upper Goulburn and Thomson water supply protection areas, respectively, supported slopes >30˚. We estimated that 89 ha of forest logged and 71 ha of forest logged in the Upper Goulburn and Thomson water supply protection areas, respectively, were >30˚ in slope. We calculated that >10% of area on slopes >30˚ had been logged in 20 cutblocks in the Upper Goulburn water supply protection area and 14 cutblocks in the Thomson water supply protection area. For our slope raster generated from the VicMap Elevation DTM (DTM), we calculated that 128 and 110 cutblocks in the Upper Goulburn and Thomson water supply protection areas, respectively, supported slopes >30˚. We calculated that 177 ha of forest logged and 118 ha of forest logged in the Upper Goulburn and Thomson water supply protection areas, respectively, were >30˚ in slope.

The smallest mean difference for the area >30˚ in slope across each cutblock was between the respective slope rasters generated using the VicMap Elevation DTM (DTM) and LiDAR 1m DEM (LiDAR 1m), where the mean difference was 0.05ha (S6 Table). The largest mean difference was 0.60 ha between the LiDAR average neighbourhood slope raster (LiDAR F5m) and the slope raster generated from the VicMap Elevation DTM (DTM). The mean difference between slope rasters generated from the SRTM DEM (SRTM) and the 1m raster (LiDAR 1m) from the LiDAR DEM, was -0.40 ha. The SRTM DEM estimated the smaller mean area.

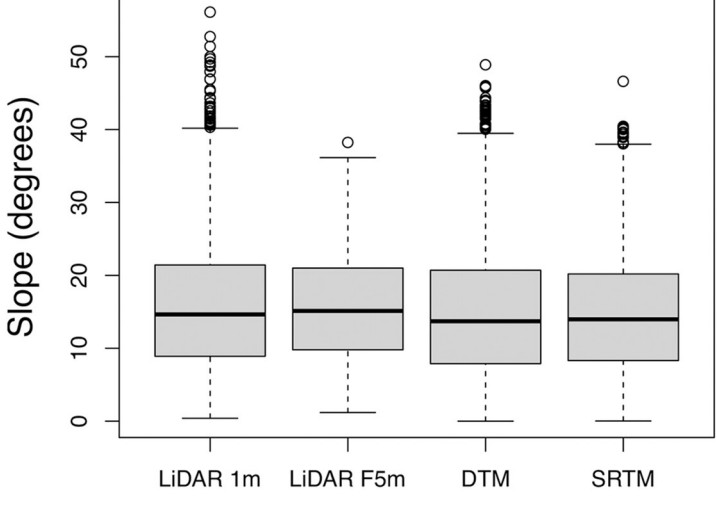

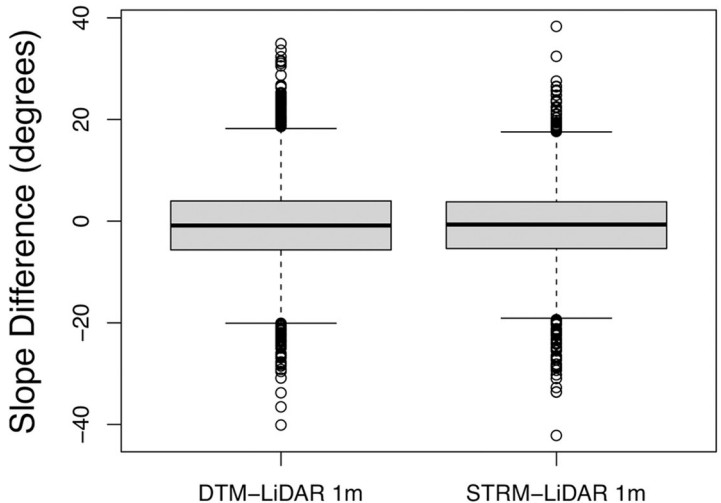

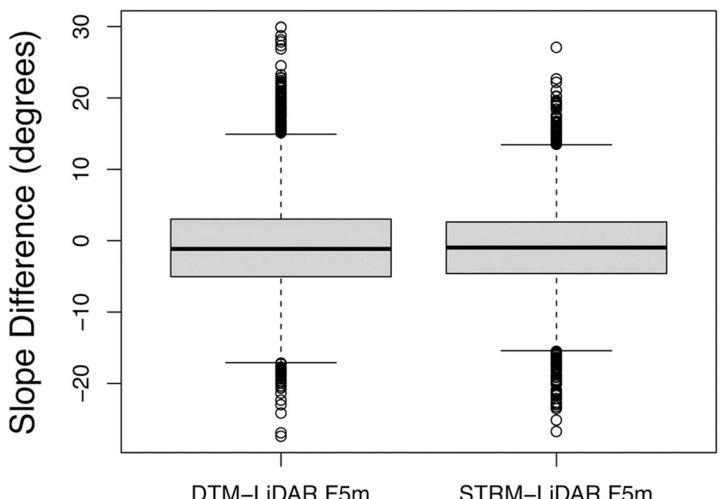

**Fig 3. Slope range in degrees for the LiDAR 1m DEM, VicMap Elevation DTM and SRTM DEM (left) and the difference in slope between LiDAR 1m DEM and the VicMap Elevation DTM (LiDAR 1m-DTM) and SRTM DEM (LiDAR 1m-SRTM).**

While the location of slopes >30˚ varied between the slope raster inclusive of all slopes generated from the LiDAR DEM (LiDAR 1m) and the slope raster generated from the VicMap Elevation DTM (DTM), the total area >30˚ calculated from the two slope rasters was not significantly different. Using the paired Wilcoxon Signed Rank test, we found that the area >30˚ in slope calculated from both DEMs to be similar (P = 0.07) (Fig 4, S7 Table). The differences between LiDAR 1m and the slope calculated from the SRTM DEM (SRTM) was significant (P<0.05), along with the difference between LiDAR F5m and the DTM slopes and the differences between the LiDAR F5m and the SRTM slopes.

## Analysis of forest blocks and cutblocks

We found that the steepest slopes logged were concentrated within a small number of forest management blocks, with Block 318 containing a significant (P>0.05) number of sites with steep slopes compared with other forest management blocks (Fig 5, S8 Table). Block 318 also contained the highest number of cut blocks with the largest of cutover areas containing slopes >30˚ for all the slope rasters analysed (Table 2) (Fig 6). The LiDAR 1m, LiDAR F5m and DTM generated slope rasters showed that all cut blocks contained slopes >30˚.

Our slope calculations from the VicMap Elevation DTM for Block 318 estimated that 81.6 ha logged was >30˚ in slope, which equated to 18.9% of the total area logged across this block. The LiDAR 1m generated slope calculated the next largest area, with 65.2 ha or 15.1% of the total area logged. The least area >30˚ in slope logged was using LiDAR F5m (average slope neighbourhood area), where 35.1 ha was >30˚ in slope, equating to 8.1% of the total area logged across the block. The SRTM generated slope raster estimated 19 of 24 cut blocks containing slopes >30˚. The extent of this area above 30˚ in slope was 37.6 ha, which equated to 8.7% of the total area logged.

**Table 1. Cutbblocks calculated with slopes >30˚ logged across the Upper Goulburn and Thomson Catchments using LiDAR at 1 m resolution (LiDAR 1m), LiDAR with a 5m (LiDAR F5m) average slope neighbourhood, the Shuttle Topography Radar Mission (STRM) at a 1 arc second resolution and the VicMap Elevation DTM (DTM) at 10m resolution.**

| Catchment | Metric | Cutblock Category | LiDAR 1m | LiDAR F5m | STRM | DTM |
|---|---|---|---|---|---|---|
| Upper Goulburn | No. of Cutblocks | All Slopes <30˚ | 2 | 24 | 63 | 26 |
| | | 0–1% Area >30˚ | 21 | 50 | 15 | 24 |
| | | 1–5% Area >30˚ | 54 | 50 | 41 | 41 |
| | | 5–10% Area >30˚ | 36 | 19 | 15 | 22 |
| | | >10% Area >30˚ | 41 | 11 | 20 | 41 |
| | | Sub Total Cutblocks >30˚ | 152 | 130 | 91 | 128 |
| Upper Goulburn | Area logged >30˚ | Subtotal (ha) | 179.0 | 78.9 | 88.9 | 177.4 |
| | | % of Total Area Logged | 7.1% | 3.1% | 3.5% | 7.0% |
| Thomson | No. of Cutblocks | All Slopes <30˚ | 0 | 29 | 81 | 34 |
| | | 0–1% Area >30˚ | 32 | 73 | 15 | 18 |
| | | 1–5% Area >30˚ | 61 | 26 | 21 | 45 |
| | | 5–10% Area >30˚ | 28 | 11 | 13 | 23 |
| | | >10% Area >30˚ | 23 | 5 | 14 | 24 |
| | | Sub Total Cutblocks >30˚ | 144 | 115 | 63 | 110 |
| Thomson | Area logged >30˚ | Subtotal (ha) | 102.4 | 37.4 | 71.1 | 118.2 |
| | | % of Total Area Logged | 4.9% | 1.8% | 3.4% | 5.7% |

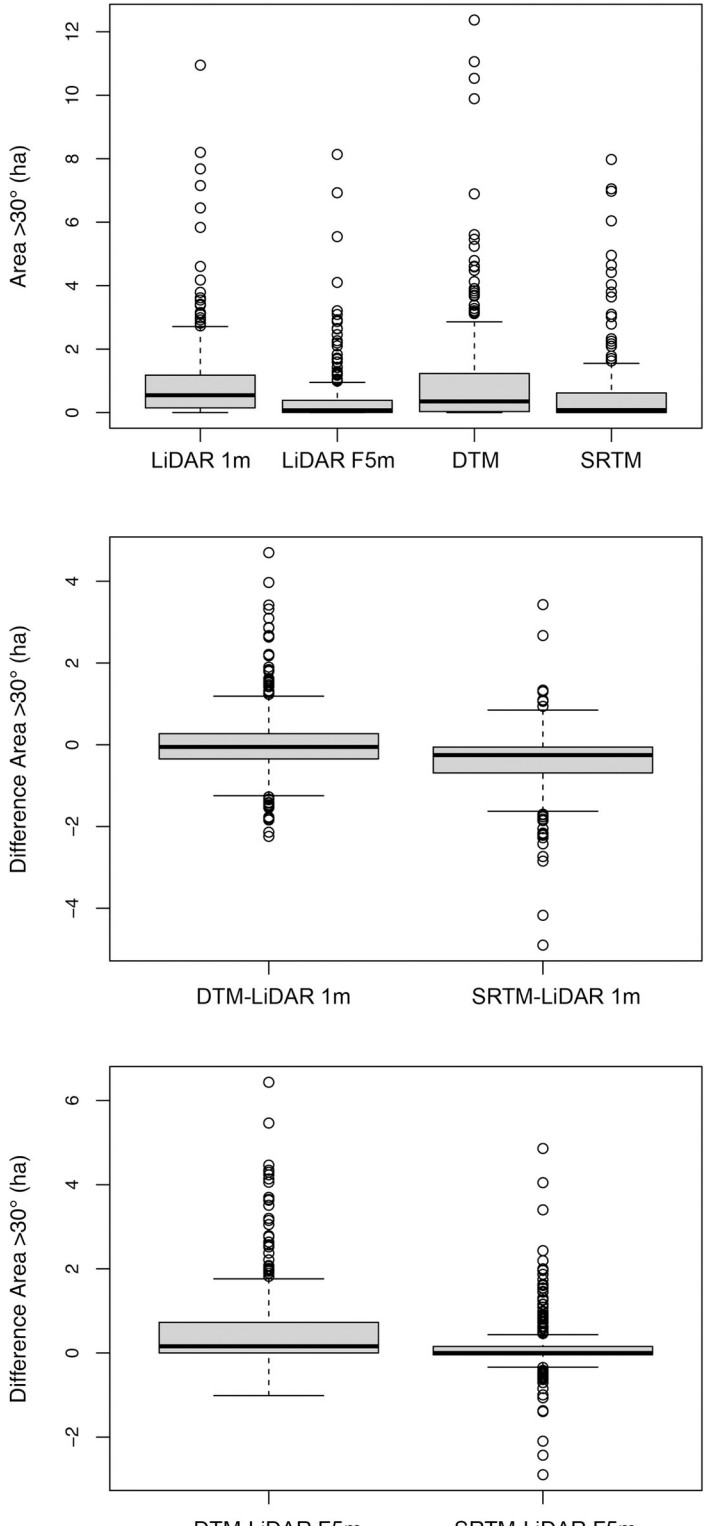

**Fig 4. Comparison between areas >30˚ across cutblocks as calculated under LiDAR 1m, LiDAR F5m, VicMap Elevation DTM and SRTM DEM slope rasters.**

## LiDAR 1m Slope

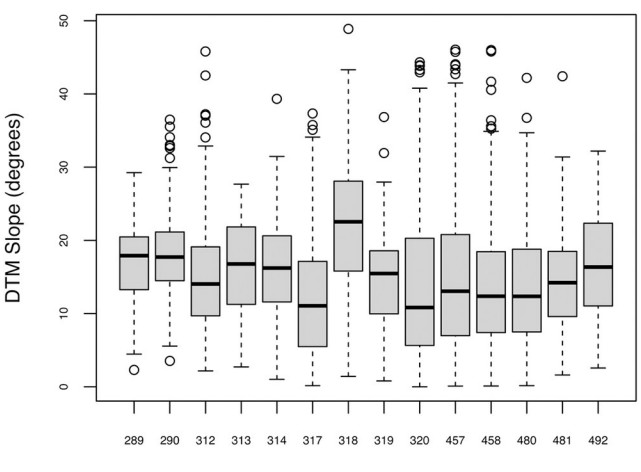

## LiDAR F5m Slope

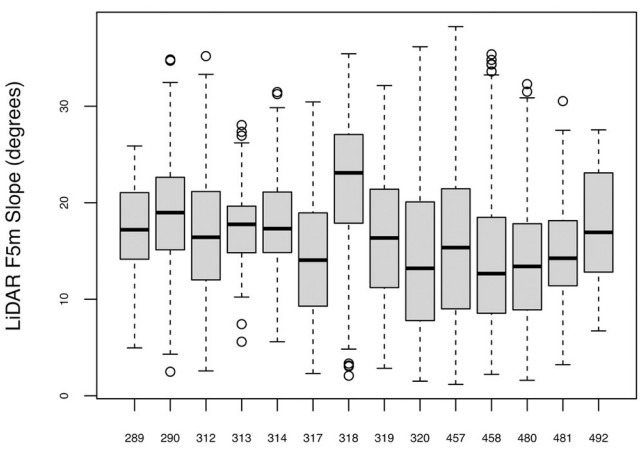

## DTM Slope

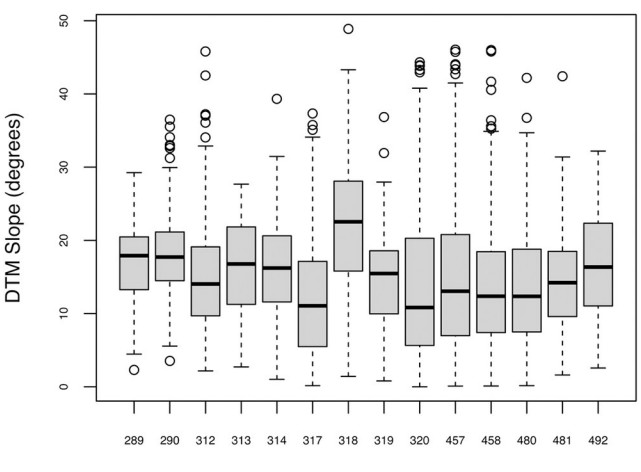

## SRTM Slope

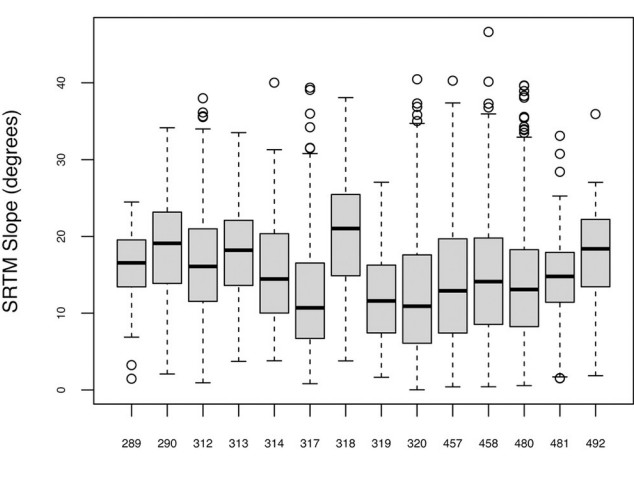

**Fig 5. Comparison of slopes logged across forest management blocks across the Upper Goulburn and Thomson water supply protection areas.**

**Table 2. Cutblocks calculated with slopes >30˚ logged across Block 318 using LiDAR at 1 m resolution, LiDAR with a 5m (LiDAR F5m) average slope neighbourhood, the Shuttle Topography Radar Mission (STRM 1 arc sec) at a 1 arc second resolution and the VicMap Elevation DTM (VicMap DTM) at 10m resolution.**

| Cutblock Category | LiDAR 1m | LiDAR F5m | DTM | STRM |
|---|---|---|---|---|
| No. of cutblocks All Slopes <30˚ | 0 | 0 | 0 | 5 |
| No. of cutblocks 0–1% Area >30˚ | 0 | 5 | 0 | 2 |
| No. of cutblocks 1–5% Area >30˚ | 3 | 10 | 2 | 6 |
| No. of cutblocks 5–10% Area >30˚ | 7 | 4 | 5 | 4 |
| No. of cutblocks >10% Area >30˚ | 14 | 5 | 17 | 7 |
| *No. of cutblocks with slopes >30° Sub Total* | *24* | *24* | *24* | *19* |
| Area logged >30˚ (ha) Subtotal | 65.2 | 35.1 | 81.6 | 37.6 |
| Area logged >30˚ % Subtotal | 15.1% | 8.1% | 18.9% | 8.7% |

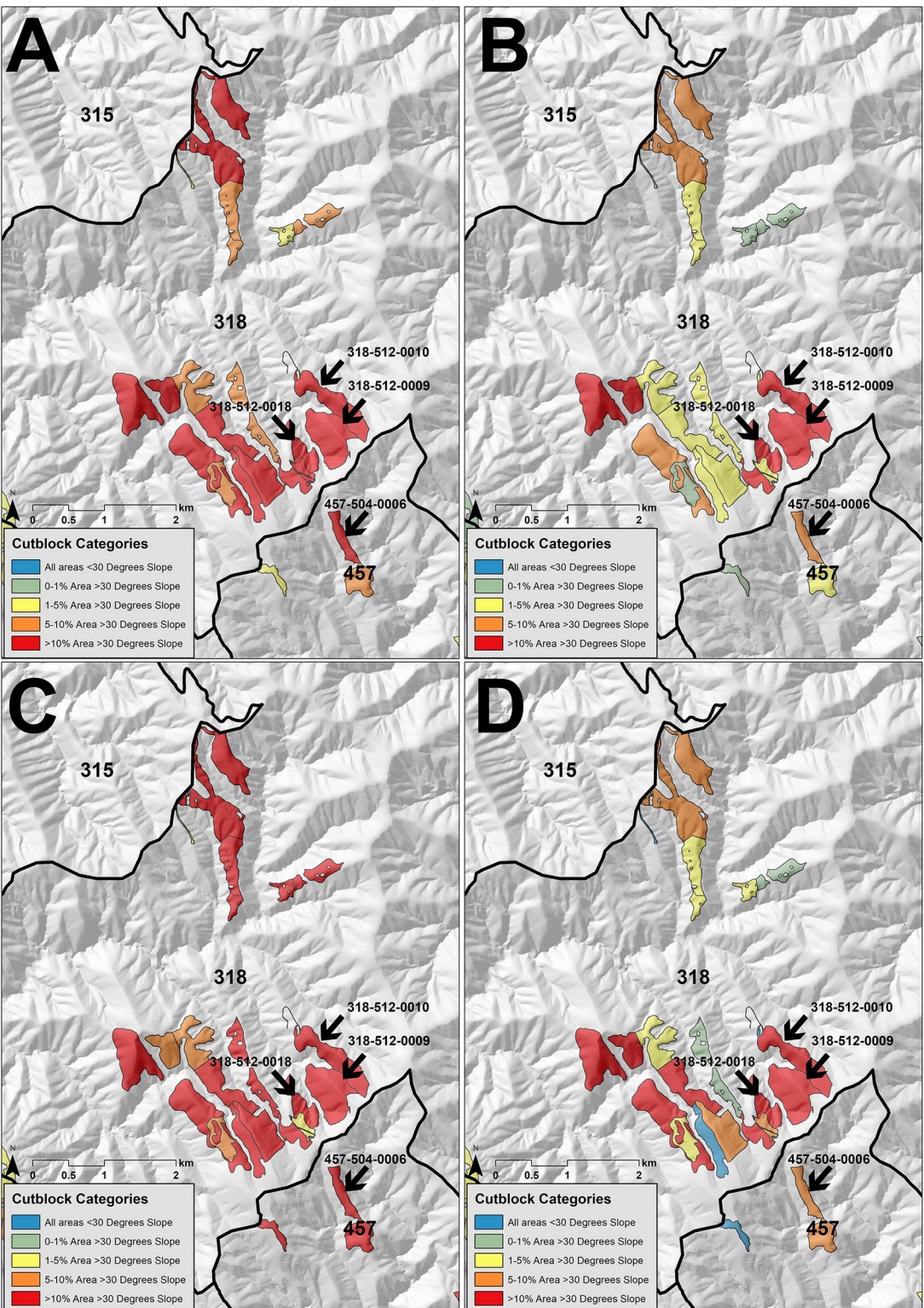

**Fig 6.** Percent of cut block area >30˚ in slope across Block 318 calculated using the LiDAR 1m DEM (A), the 5 m average slope calculated using the LiDAR DEM (LiDAR F5m) (B), calculated using the VicMap Elevation DTM (C) and calculated using the SRTM DEM (D).

## Slope measurements across sample cut blocks and DEM/DTM slope comparisons

Slopes across a sample of cutblocks were measured across 62 transects by one of us (CT) (26 transects) and the Timber Harvest Compliance Unit of the OCR (36 transects) [22,23]. This consisted of Cutblocks 318-512-0009, 318-512-0010, 320-501-0024 and 457-504-0006. For Cutblock 318-512-0009 (within Forest Block 318), we measured slopes across 11 transects, ranging between 29.6˚ and 42.8˚ (Fig 7, S9 Table). Transect distances ranged between 10m and 34.5m. Our LiDAR slope calculations had the strongest alignment with in-field measurements. Of the 11 transects we measured for Cutblock 318-512-0009, seven were within ±2˚ of our LiDAR average slope calculations (S11 Table). Our slope calculations generated from the SRTM DEM under-estimated on-site measurements on Cutblock 318-512-0009, with 10 of the 11 average slope calculations for the transects being less than on-site measurements. Average slope calculations generated from the VicMap Elevation DTM were below the measured slope value for only three of the transects. We measured six transects at Cutblock 318-512-0018. These transects ranged between 30˚ and 37.4˚ in slope and transect distances ranged between 12.5m to 32.5m in length (S3 Fig, S9 Table). The LiDAR average calculations were within ±2˚ of on-site slope measurements for five transects. The VicMap Elevation DTM and the SRTM DEM under predicted the slopes for this cutblock for three and five transects, respectively. We measured nine transects in Cutblock 457-504-0006, which is adjacent to Block 318. These transects ranged between 29˚ and 33.8˚ and transect distances ranged between 12m to 31.5m in length (S9 Table, S4 Fig). Of these, five were within ±2˚ of our LiDAR average slope calculations (S9 Table). Both the VicMap Elevation DTM and SRTM DEM under-predicted the slopes for seven and all the transects on Cutblock 457-506-0006, respectively.

Cutblock 318-512-0010 (within Forest Block 318) was investigated by the Timber Harvest Compliance Unit of the OCR (Fig 7), where they measured 15 transects. These ranged between 27.8˚ and 33.3˚ in slope and where between 30.5m and 102.6m in length. Of the transects measured, 13 were within ±2˚ of our LiDAR average slope calculations (S9 Table). The SRTM DEM under predicted 13 transects and the VicMap Elevation DTM overpredicted the slope for all transects. Cutblock 320-501-0024 was also investigated by the Timber Harvest Compliance Unit (Fig 8), where they measured 21 transects. These ranged between 24.5˚ and 38˚ in slope and where between 14.8m and 79.6m in length. Of the transects measured, 17 of these were within ±2˚ of our LiDAR average slope calculations (S9 Table). The SRTM DEM under predicted 16 transects and the VicMap Elevation DTM under predicted 14 transects.

As the calculated slopes data followed a normal distribution and the measured data did not, we conducted both the paired Student's T Test and the paired Wilcox signed rank test to compare the measurements and slope calculations generated from the DEMs and DTM for all transects surveyed. Both tests showed that both the slope measurements and the LiDAR calculations for the transects were similar (P = 0.4 for Wilcox paired signed rank test and P = 0.11 for Student's paired T test) (S10 and S11 Tables). Variation was the least as indicated by the lowest of standard deviations (2.23˚) (S12 Table). The slope calculations generated from the VicMap Elevation DTM for the transects were similar to the on-site measurements (P = 0.80 for the paired Wilcox signed rank test and P = 0.86 for the paired Student's T test) (S10 and S11 Tables). This result indicated that the means and medians of the measurements and the slope calculations were similar. Our slope calculations generated from the VicMap Elevation DTM neither under-estimated nor over-estimated onsite measurements across the transects (Fig 9). However, the standard deviation between the measured slope values and the slope calculations generated from the VicMap Elevation DTM was highest of the comparisons at 7.5˚ in slope (S12 Table), which revealed variation between the on-site measurements and

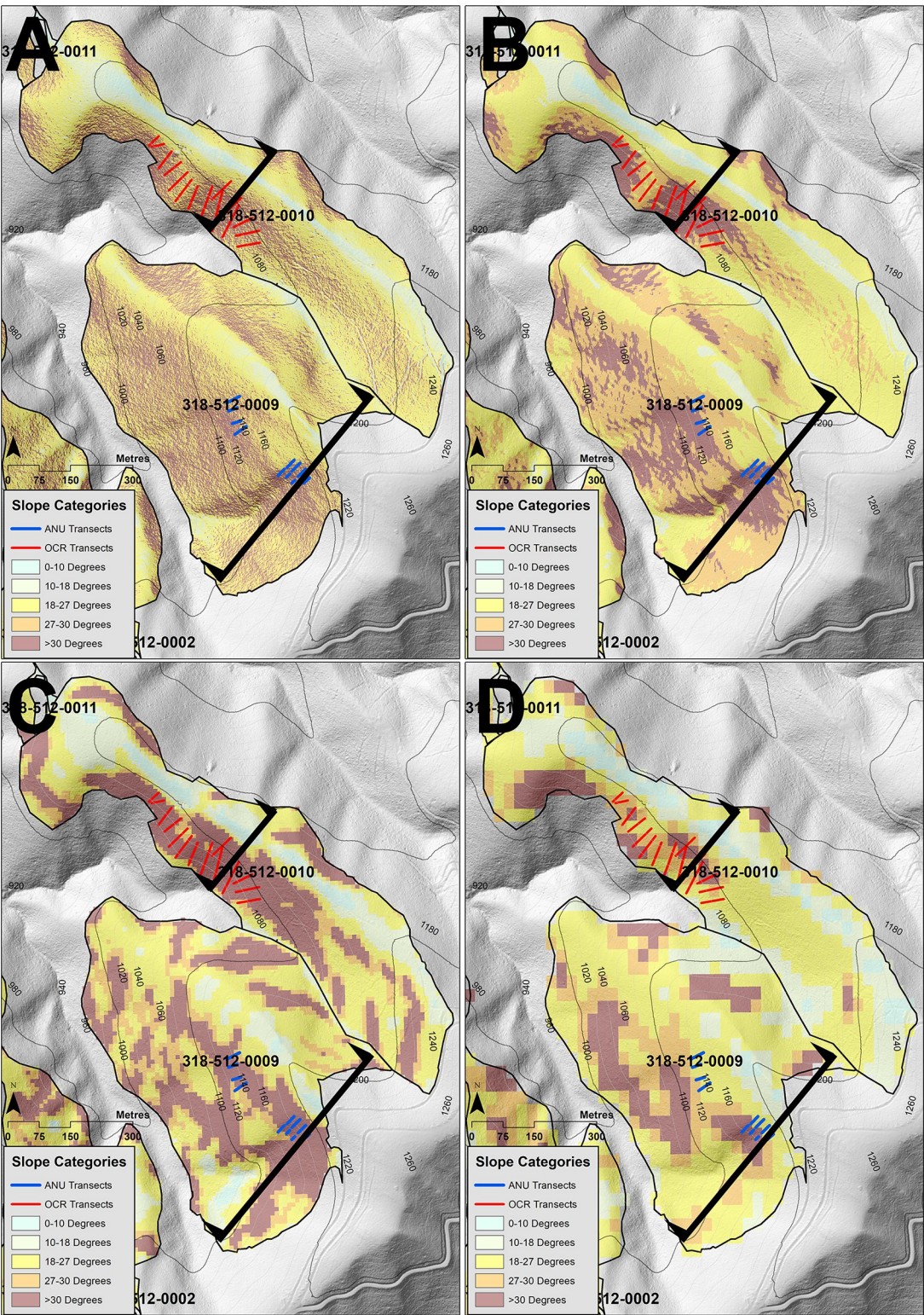

**Fig 7. Comparison of slope calculations for Cutblock 318-512-0009 and 318-512-0010.** LiDAR derived slope at 1m resolution (Fig 7A); LiDAR derived slope with an average slope neighbourhood radius of 5m (Fig 7B); STRM derived slope at 1 arc second resolution (Fig 7C); DTM derived slope at 10m resolution (Fig 7D). Blue and Red lines show measured transects for the ANU and OCR Transects, respectively. Slope cross sections A and B are shown.

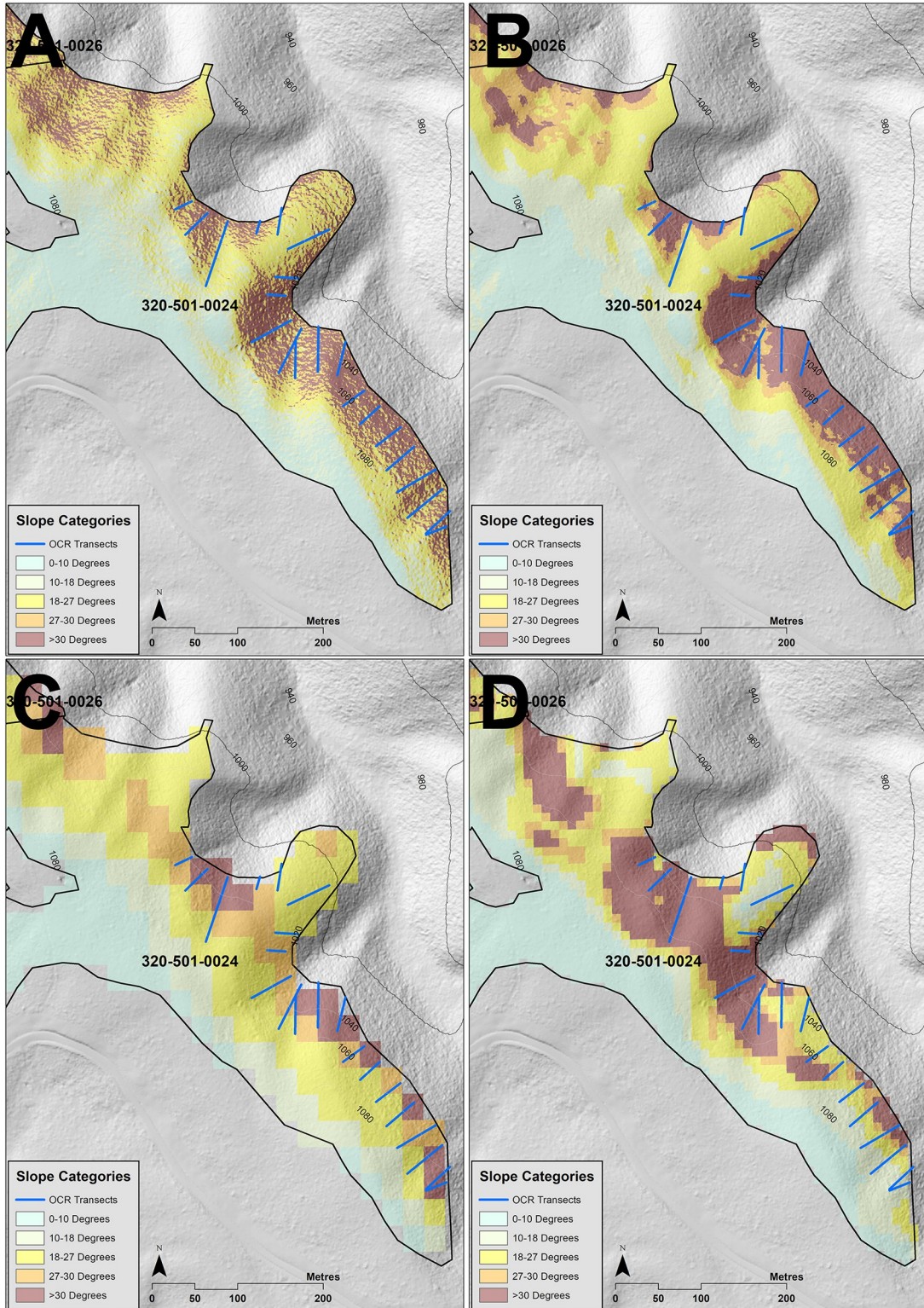

**Fig 8. Comparison slope calculations with transects measured on Cutblock 320-501-0024 by the Timber Harvest Compliance Unit of the OCR.** LiDAR derived slope at 1m resolution (Fig 8A); LiDAR derived slope with an average slope neighbourhood radius of 5m (Fig 8B); STRM derived slope at 1 arc second resolution (Fig 8C); DTM derived slope at 10m resolution (Fig 8D).

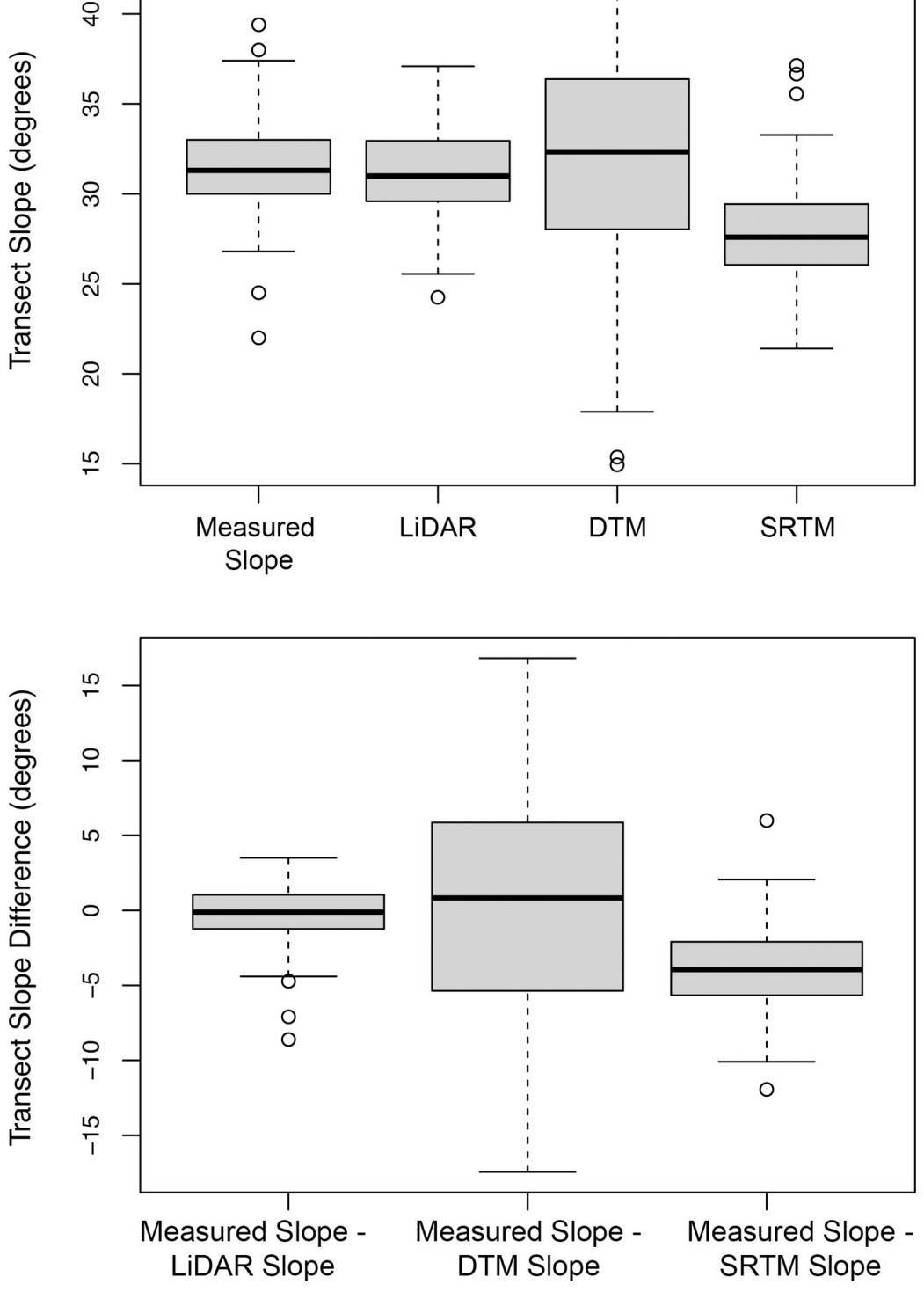

**Fig 9. Boxplots of transect measurements compared with slope rasters derived from LiDAR 1m Dem, VicMap Elevation DTM and SRTM DEM.**

**Table 3. Sample of cutblocks calculated with slopes >30˚ logged using LiDAR at 1 m resolution (LiDAR 1m), LiDAR with a 5m (LiDAR F5m) average slope neighbourhood, the Shuttle Topography Radar Mission (STRM) at a 1 arc second resolution, and the VicMap Elevation DTM (DTM) at 10m resolution.** These cutblocks were measured by the authors and also by staff from the OCR.

| Catchment | Cutblock Address | LiDAR 1m Area >30˚ (ha) | LiDAR F5m Area >30˚ (ha) | VicMap DTM Area >30˚ (ha) | SRTM Area >30˚ (ha) | Total Area (ha) Logged |
|---|---|---|---|---|---|---|
| Upper Goulburn | 318-512-0009 | 10.95 | 8.14 | 12.37 | 6.04 | 34.90 |
| | 318-512-0010 | 5.83 | 4.10 | 10.53 | 3.10 | 29.57 |
| | 318-512-0018 | 3.55 | 2.26 | 3.84 | 2.17 | 21.52 |
| | 320-501-0024 | 2.71 | 2.46 | 3.12 | 1.08 | 28.93 |
| Thomson | 457-504-0006 | 1.37 | 1.00 | 3.21 | 0.93 | 10.12 |

slope calculations. The slope mean and median calculated from the SRTM DEM was lower than medians for the transects (P<0.05), demonstrating that slope calculations derived from this DEM underestimated the slopes of the measured transects.

## Slope calculations across sample cut blocks

Across sample Cutblocks 318-512-0009, 318-512-0010, 320-501-0024 and 457-504-0006, we found the SRTM DEM predicted the smallest areas >30˚ in slope, the VicMap Elevation DTM the largest areas >30˚, and the LiDAR generated slope area calculations in between the SRTM DEM and the VicMap Elevation DTM slope calculations (Table 3). For Cutblock 318-512-0009 in Forest Management Block 318, the logged area was 34.9 ha and slope areas >30˚ were calculated to be 10.95 ha, 8.14 ha, 12.37 ha and 6.04 ha for LiDAR 1m, LiDAR F5m, DTM and SRTM slope rasters, respectively (Fig 7, Table 3). Differences in slope were a result of differences in the elevation predicted LiDAR 1m DEM, VicMap Elevation DTM and the SRTM DEM. For our sample slope profile of Cutblock 318-512-0009 (Fig 10), we found that the difference between the LiDAR DEM and the SRTM DEM in elevation at the ridgeline was 9m, but the difference in the valley was 33m, with the SRTM DEM predicting higher elevations for both ridgeline and valley. This difference lessened the slope degree, thereby underestimating the area of slope when compared to our LiDAR slope calculations. As a result, the area logged >30˚ estimated for Cutblock 318-512-0009 from the SRTM DEM was less than both the LiDAR 1m calculations and the LiDAR F5m average slope calculations. The peak of the cutblock as predicted by the VicMap Elevation DTM was only 1.47m higher than the peak

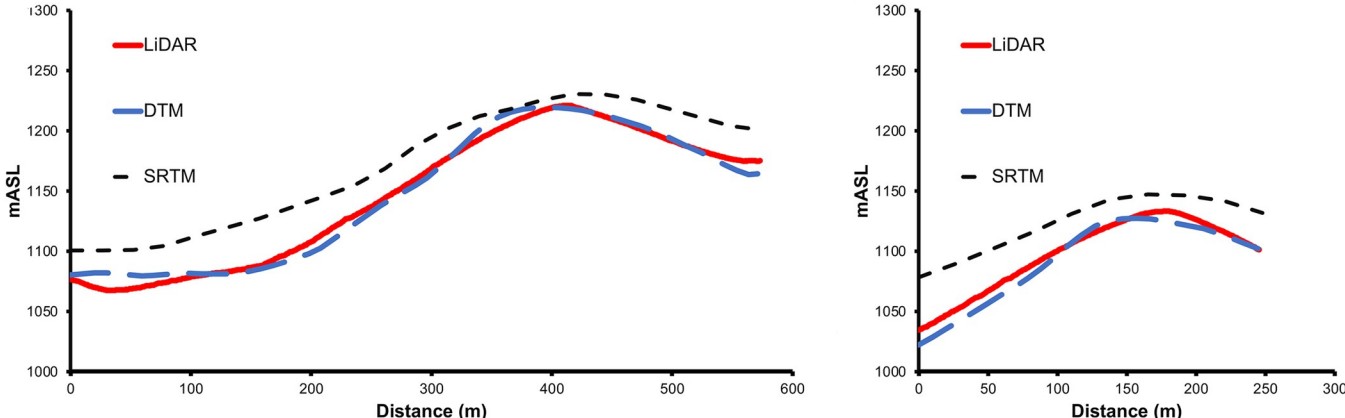

**Fig 10.** Slope comparisons along transects across the area of Cutblock 318-512-0009 (left) and the Cutblock 318-512-0010 (right).

predicted by the LiDAR DEM ridge, but its location was 16m further towards the valley, thereby predicting a steeper slope for this sample transect.

For Cutblock 318-512-0010, differences in slope area >30˚ were also a result of differences in the elevation predicted LiDAR 1m DEM, VicMap Elevation DTM and SRTM DEM. The STRM DEM over-estimated the ridgeline and the valley by 14m and 43m, respectively, when compared with the LiDAR DEM. This lessened the slope gradient compared with the LiDAR 1m DEM. The SRTM DEM slope calculations estimated 3.1 ha >30˚ in slope, whereas the LiDAR 1m and LiDAR F5m calculations estimated 5.8 ha and 4.1 ha >30˚ in slope logged, respectively (Fig 8, Table 3). The VicMap Elevation DTM under predicted the elevation by 19m and 12m for the peak and the valley, respectively, compared with the LiDAR DEM (Fig 10). However, it predicted a steeper slope compared with the LiDAR DEM because the ridge-line was 21.9m closer to the valley compared with the LiDAR DEM predicted ridgeline.

## Discussion

The information gained through the use of DEMs can greatly assist forest management planning [61]. Indeed, DEMs have become a common method for extracting topographical information [51]. However, the accuracy of resolution of different DEMs can vary, with lower resolution DEMs potentially underestimating the area of steep slopes across forested regions [19]. This means that different DEMs may provide different results about the location and extent of different kinds of terrain in a given area. In the particular context of this study, and during correspondence with the OCR, that agency claimed that slope calculations derived from the SRTM DEM and the VicMap Elevation DTM were unreliable and often overstated the degree of slope when compared to in-field measurements [23]. We sought to test this assumption by answering two key questions: *What are the levels of congruence among slope calculations derived from multiple Digital Elevation Models*? and, *How do these calculations compare with on-site slope measurements*?

We found that whilst different DEMs varied across our study sites, the claim by the OCR that the SRTM DEM and the VicMap Elevation DTM overstated the degree of slope when compared to in-field measurements could not be substantiated. In our analysis, we found that slope calculations generated from the VicMap Elevation DTM were variable, and slope calculations generated from the SRTM DEM under-estimated slope steepness when compared to in-field measurements. As expected, we found LiDAR to provide the most accurate match to in-field measurements. Whilst there was variation between the slope raster generated from the LiDAR DEM and the slope raster generated from the VicMap Elevation DTM in terms of the location of slopes >30˚ across cutblocks, we found that the area estimates matched in terms of the total area logged on slopes >30˚.

### Comparisons between digital elevation models

Differences between DEMs have been extensively analysed in previous studies [19,21,59]. For example, DeWitt et al. [21] found that the SRTM DEM over-predicted elevation compared with a LiDAR-generated DEM. These differences can be explained by positional errors that exist in datasets. For example, the LiDAR dataset we used has a positional accuracy of +/-0.2m [38], which is significantly greater than that of the VicMap Elevation DTM, which is +/-12.5m horizontally and +/-5m vertically [25]. The report by Gallant et al. [40] found elevation differences between the VicMap Elevation DTM [25] and the SRTM DEM across our study region, which was likely the result of vegetation having not been sufficiently removed or treated in the generation of the SRTM DEM. Reasons for this included areas of riparian and remnant vegetation not being adequately mapped and therefore difficult to remove from the SRTM DEM,

forested mountainous areas where the vegetation offset had been systematically under-esti-mated, and areas of more generalized valley prediction in steep dissected valleys.

Other studies have compared the SRTM DEM to LIDAR in other regions of the world. For example, DeWitt et al. [21] compared several regional and global-scale DEMs, including the SRTM DEM, to a high-accuracy LiDAR DEM to assess their differences quantitatively in rug-ged topography in an area of Utah in the USA. That study ranked the SRTM DEM as having a low accuracy with a RMSE value of 14.90 m. Their Ordinary Least Squares regression provided insights about the complex relationships between elevation, slope, and differences between DEMs, suggesting that the SRTMDEM predicted a topographic surface substantially higher in elevation than estimated by the LiDAR DEM. The study by Gonga-Saholiariliva et al. [59] found that the elevation difference between the SRTM DEM and a reference LiDAR DEM was not random, but spatially dependent. Large areas of difference were spatially concentrated in valleys. These factors have implications for the estimation of slopes across the landscape. In a study in Kalimantan, Putz et al. [19] found that lower resolution DEMs (similar to the SRTM DEM (~1 arc second) we employed) underestimated the area of steep slopes compared to the slope calculations derived from LiDAR. Lower resolution DEMs can smooth the landscape by averaging the contents of a cell. This can have the effect of decreasing predictions of steep slopes, such as in gullies [19].

## Comparisons with previous analyses

We first analysed the Upper Goulburn and Thomson water supply protection areas in a sub-mission to the audit of logging operations across these catchments against the Forest Steward-ship Council (FSC) Controlled Wood Standard for Forest Management Enterprises (FSC-STD-30-010 V2.0) [20]. At the time of the audit, only the SRTM and the VicMap Eleva-tion DTM 10m datasets were publicly available. We calculated that 75% of cut blocks across the Upper Goulburn water supply protection area contained slopes >30˚, with 15% of those cut blocks containing >10% of the net areas >30˚ in slope [20]. Our subsequent analysis using LiDAR found 152 out of 154 cut blocks supported slopes >30˚ with 27% of cut blocks contain-ing >10% of the net areas >30˚ in slope. However, these calculations are inclusive of smaller areas (some areas ~1m$^2$). By applying the 5 metre average slope area neighbourhood (LiDAR F5m), we calculated that 84% of cut blocks contained slopes >30˚ with 7.1% of cut blocks con-taining >10% of their net areas logged >30˚ in slope. Our analysis presented here confirmed the findings of Taylor and Lindanmayer [20] that forests on slopes >30˚ were logged across the Upper Goulburn water supply protection area and, that occurrences of such breaches of codes of practice were widespread. The differences between the different datasets are the loca-tions of the steep slopes. This is due to positional accuracies within each of the datasets [21].

## Comparisons with analysis of steep slopes by VicForests

In a presentation to the Environment and Communications Legislation Committee Inquiry into the Environment Protection and Biodiversity Conservation Amendment (Regional Forest Agreements) Bill 2020, Dawson [62] claimed that VicForests had undertaken an analysis of slopes >30˚ logged in the Upper Goulburn water supply protection area, where it had used LiDAR (DELWP 2019) and the DEMs used in Taylor and Lindenmayer [20] and Taylor and Lindenmayer [57,63]. In their analysis, VicForests [63] claimed they calculated only 2% of the total net area logged since 2004 in the Upper Goulburn water supply protection area consisting of slopes >30˚. This was substantially smaller than the logged area >30˚ in slope that we calcu-lated for the Upper Goulburn water supply protection area, which was 7.1% or 179 ha. VicFor-ests [63] claimed its slope calculations included small patches down to 1 square metre and

these did not constitute a breach of the Management Standards and Procedures. However, the claim made by VicForests that its slope analysis included small areas (~1m$^2$ >30˚) could not be substantiated given that VicForests decreased the resolution of its LiDAR analysis to a 10m slope raster [57] and that any area calculated to be >30˚ would be an average slope for a 100m$^2$ (10m x 10m) area.

VicForests also calculated that 10 cutblocks contained >10% of net area logged >30˚ in slope. This included Cutblock 318-512-0009, where VicForests calculated 7.96 ha or 22.3% of net area logged was >30˚ in slope, and Cutblock 318-512-0010, where VicForests calculated 3.97 ha or 13.4% of net area logged was >30˚ in slope. Compared with our LiDAR F5m average neighbourhood slope analysis, we calculated 8.14 ha and 4.10 ha for these cut blocks, respectively. The total area logged as calculated by VicForests in the Upper Goulburn water supply protection area to be >30˚ across the Forest Management Blocks analysed in our study was 81 ha [63], which was comparatively close to the slope area >30˚ estimated generated under the LIDAR F5m average slope (79 ha) (Table 1).

## Implications for forest management

We suggest that the use of DEMs is of considerable value in proactive forest planning. Such datasets can help identify areas which must be excluded from future logging operations and assess compliance with regulatory codes of practice. DEMs generated from high resolution LiDAR data are a cost-effective method of obtaining accurate data and providing greater transparency to stakeholders. Allowing stakeholder access to such data for repeat analyses or new analysis increases the social license for forestry enterprises. For compliance audits, assessors can examine entire landscapes and identify areas that may present a risk of non-compliance. This avoids the potential shortcomings of field sampling, including where previous audits have missed areas where non-compliance has occurred.

While LiDAR provides a high resolution and high accuracy dataset, it is not currently available for all areas. This is the case for large forest parts of Victoria outside of the Victorian Government's LiDAR project for the Central Highlands [38]. Furthermore, some LiDAR datasets may not be publicly accessible; the dataset used in our study was only made available through an FOI request following an audit of VicForests' logging operations under the FSC Controlled Wood Standard for Forest Management Enterprises (FSC-STD-30-010 V2.0). Where LiDAR data are unavailable, lower resolution datasets, such as the SRTM DEM and the VicMap Elevation DTM, are also useful tools for locating slopes that present concerns for forest management and compliance. While comparative analyses have shown that DEMs of lower resolution can underestimate the extent of areas on steep slopes [19], in this study we have shown that they are still able to indicate where steep slopes are likely to occur. Using LiDAR data allows more precise analysis of the extent and locality of steep slopes. Differences between DEMs can be variable but nevertheless important where compliance with laws preventing logging on steep slopes is required. This is particularly critical in water catchment areas where logging of slopes >30˚ increases the likelihood of erosion threatening the provision of water fit for human consumption.

## Conclusions

Our work demonstrates that LiDAR, DEMs, and DTMs can be valuable in assisting in environmental management and helping with compliance with forest laws and codes of practice. However, it is essential to be clear about which method/s are being used so that assessments are transparent to stakeholders. We have provided an analysis of forest terrain in the Upper Goulburn and Thomson water supply protection areas which have been previously logged.

Our results have shown that logging operations in these areas have not always been compliant with existing forest laws and codes of practice.

The DEMs and DTMs examined in this study also can be used in work on future forest planning, allowing for more accurate identification of areas which should be exempt from logging. We have shown that areas >30° in slope were logged across the Upper Goulburn and Thomson water supply protection areas and it is critical these steep slopes are excluded from further logging to protect essential ecosystem services and to comply with regulations. While some areas of slopes >30° were mapped using the VicMap Elevation DTM, it is unclear whether the DEMs and DTMs were used to map steep slopes in forests allocated to logging across the wider landscape. However, we suggest there is considerable value in using DEMs and DTMs in proactive forest planning and to prevent logging in inappropriate areas. The use of these models will allow resource managers to more accurately identify areas which must be excluded from logging and, in turn, recalculate timber yields in light of these exclusions.

## Supporting information

**S1 Fig. QQ Plots assessing for normalcy for Elevation and calculated Slope data.**
(DOCX)

**S2 Fig. QQ Plots assessing for normalcy for Cut Block Area >30° in Slope.**
(DOCX)

**S3 Fig. Comparison slope calculations with transects measured on cut block 318-512-0018 by the Authors.** LiDAR derived slope at 1m resolution (S3A Fig); LiDAR derived slope with an average slope neighbourhood radius of 5m (S3B Fig); STRM derived slope at 1 arc second resolution (S3C Fig); DTM derived slope at 10m resolution (S3D Fig).
(DOCX)

**S4 Fig. Comparison slope calculations with transects measured on cut block 457-504-0006 for this study.** LiDAR derived slope at 1m resolution (S4A Fig); LiDAR derived slope with an average slope neighbourhood radius of 5m (S4B Fig); STRM derived slope at 1 arc second resolution (S4C Fig); DTM derived slope at 10m resolution (S4D Fig).
(DOCX)

**S5 Fig. QQ Plots assessing for normalcy across Transects measured for this study (ANU Transects) and by the Timber Harvest Compliance Unit of the OCR (OCR Transects).**
(DOCX)

**S1 Table. Moran's I Spatial Autocorrelation for the analysis grid points across cut blocks in the Upper Goulburn and Thomson water supply projection areas.**
(DOCX)

**S2 Table. Test for Normalcy using the Shapiro-Wilks Test and QQPlots for the Elevation, Slope, Cut Block Area >30° and the Transects measured on site (ANU and OCR).**
(DOCX)

**S3 Table. Descriptive Statistics for the Elevation (mASL) and difference between the LiDAR 1m DEM and the VicMap Elevation DTM and SRTM DEM (m).**
(DOCX)

**S4 Table. Moran's I Spatial Autocorrelation for the elevation difference between LiDAR analysis grid points and the VicMap Elevation DTM and SRTM DEM across cut blocks in the Upper Goulburn and Thomson water supply projection areas.**
(DOCX)

**S5 Table. Descriptive Statistics for the Slope (mASL) and difference between the LiDAR 1m DEM and the VicMap Elevation DTM and SRTM DEM (m).**
(DOCX)

**S6 Table. Ordinary Least Squares test between the slope rasters generated from the LiDAR 1m DEM, including the LiDAR 1m and LiDAR F5m slope rasters, and the VicMap Elevation DTM and SRTM DEM.**
(DOCX)

**S7 Table. Moran's I Spatial Autocorrelation for the slope difference between LiDAR analysis grid points and the VicMap Elevation DTM and SRTM DEM across cut blocks in the Upper Goulburn and Thomson water supply projection areas.**
(DOCX)

**S8 Table. Descriptive Statistics for the area >30˚ logged in each cut block and difference between the LiDAR 1m DEM and the VicMap Elevation DTM and SRTM DEM (ha).**
(DOCX)

**S9 Table. Wilcoxon signed rank test comparing areas >30˚ in slope logged calculated by the LiDAR 1m, LiDAR F5m, VicMap Elevation DTM and the SRTM DEM.**
(DOCX)

**S10 Table. ANOVA and Tukey's HSD Test between Block 318 and other forest management blocks.**
(DOCX)

**S11 Table. Transects measured by the authors (ANU Transects).**
(DOCX)

**S12 Table. Transects measured by the Timber Harvest Compliance Unit of the Office of the OCR (OCR Transects).**
(DOCX)

**S13 Table. Wilcoxon signed rank test comparing measured slope to the average slope calculations derived from the LiDAR 1m DEM, the VicMap Elevation DTM and the SRTM DEM.** Adjusted $R^2$ Values were generated from an Ordinary Least Squares test.
(DOCX)

**S14 Table. Descriptive statistics of the ANU and OCR Transects covering measured slope and the average slope calculations generated from the LiDAR 1m DEM, the VicMap Elevation DTM and the SRTM DEM.**
(DOCX)

**S15 Table. Cut blocks calculated with slopes >30˚ logged across the Upper Goulburn water supply protection area as calculated by VicForests using a resampled LiDAR DEM at 10m resolution.**
(DOCX)

## Acknowledgments

We acknowledge the GunaiKurnai and Taungurung People upon whose Country we conducted our analysis. We sincerely acknowledge their Elders, past, present and emerging, and their continuing Custodianship of their Country. We thank Ms M. Dawson for commentary

that stimulated the analyses which underpinned this paper. We are grateful to VicForests for providing the LiDAR data used in our spatial analyses. Ms. T. Boyer and Dr. J. Williams assisted with editorial aspects of manuscript preparation. We thank Professor M. Hutchinson for insightful discussions on the accuracy of terrain analysis and whose advanced spatial mathematical routines were instrumental in the development of digital elevation and digital terrain models.

## Author Contributions

**Conceptualization:** Chris Taylor, David B. Lindenmayer.

**Data curation:** Chris Taylor.

**Investigation:** Chris Taylor.

**Methodology:** Chris Taylor.

**Resources:** David B. Lindenmayer.

**Supervision:** Chris Taylor, David B. Lindenmayer.

**Visualization:** Chris Taylor.

**Writing – original draft:** Chris Taylor, David B. Lindenmayer.

**Writing – review & editing:** Chris Taylor, David B. Lindenmayer.

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
