## [Decision Letter · Decision Letter 0]

9 Feb 2022

PONE-D-21-34995The use of spatial data and satellite information in legal compliance and planning in forest managementPLOS ONE

Dear Dr. Lindenmayer,

Thank you for submitting your manuscript to PLOS ONE. After careful consideration, we feel that it has merit but does not fully meet PLOS ONE’s publication criteria as it currently stands. Therefore, we invite you to submit a revised version of the manuscript that addresses the points raised during the review process.

ACADEMIC EDITOR: please make the framing of the work more clearer and clarify the data used.Please ensure that your decision is justified on PLOS ONE’s publication criteria and not, for example, on novelty or perceived impact.

We look forward to receiving your revised manuscript.

Kind regards,

RunGuo Zang

Academic Editor

PLOS ONE

Journal Requirements:

2. We note that Figures 1, 6, 7 and 8 in your submission contain map images which may be copyrighted. All PLOS content is published under the Creative Commons Attribution License (CC BY 4.0), which means that the manuscript, images, and Supporting Information files will be freely available online, and any third party is permitted to access, download, copy, distribute, and use these materials in any way, even commercially, with proper attribution. For these reasons, we cannot publish previously copyrighted maps or satellite images created using proprietary data, such as Google software (Google Maps, Street View, and Earth). For more information, see our copyright guidelines: http://journals.plos.org/plosone/s/licenses-and-copyright.

1. You may seek permission from the original copyright holder of Figure 1, 6, 7 and 8 to publish the content specifically under the CC BY 4.0 license.  

Additional Editor Comments (if provided):

please revise the ms according to the concerns of the reviewers.

Reviewers' comments:

Reviewer's Responses to Questions

**Comments to the Author**

1. Is the manuscript technically sound, and do the data support the conclusions?

Reviewer #1: Yes

Reviewer #2: Yes

2. Has the statistical analysis been performed appropriately and rigorously? 

Reviewer #1: Yes

Reviewer #2: Yes

3. Have the authors made all data underlying the findings in their manuscript fully available?

Reviewer #1: Yes

Reviewer #2: Yes

4. Is the manuscript presented in an intelligible fashion and written in standard English?

Reviewer #1: Yes

Reviewer #2: Yes

5. Review Comments to the Author

Reviewer #1: This is an interesting paper which presents a very detailed applied study concerning the application of digital terrain analysis, specifically topographic slope angle, to forest management prescriptions. The results are novel in terms of the context, namely, the efficacy of differently grained DEMs and the slope derivative to inform the monitoring and evaluation of forest management prescriptions designed to protect water catchments. The paper presents an extremely thorough empirical analysis as well as providing an in-depth discussion of the practical relevance of the results. The paper warrants publishing in PLOS ONE subject to a minor revision to take account of the following points.

L60 - “isn’t” should be “is not”

L62 – remove brackets as this is a key point, not an aside

L72 does “best” = “most cost effective”? What are criteria for assessing what is “best” here?

L74 – strictly speaking a DEM is not remote sensing data. It is a spatial data layer modelled from a satellite-based or otherwise air-borne active sensor.

L77-78 – this sentence needs to be rewritten to make clearer the meaning given it is the basis for the relevance of the paper’s analyses.

L 105-108, 109-112 – these read more like conclusions, rather than introductory statements, as they are anticipating the results.

L 192 – how was slope calculated in Arc GIS? I.e. different functions can be used to calculate gradients in the X and Y direction.

L221 – can the authors clarify that there were Lidar shots for every grid cell? If not, then some interpolation would have been required.

L224 – was there a rationale behind the class intervals?

L268 – it is not clear why you would need to test for spatial autocorrelations given all the data being analysed are DEMs on a regular grid, it is not clear why autocorrelation would be an issue?? What was the autocorrelation of concern?

L346 – I think the differences between the values of the differently scaled slope grides are better described as differences and not errors. I say this because slope can be accurately calculated at a range of scales for the same location and differ in their values.

L360 - clarify that this result was found from all three scaled slope grids, or if not, which scale was used.

General comments:

1. It would be helpful if the authors could clarify if all three scales are relevant to the scale at which the forest management guidelines are intended to be applied. For example, I don’t think that forest management prescriptions are meant to be applied at a 1m resolution. Wouldn’t the appropriate scale of analysis be the slope calculated at a scale that captures the variation within a logged coup; which I assume is equivalent to what the authors call a cut block? In any case, some discussion is warranted of the scale at which slope is calculated compared with the scale at which forest management prescriptions are applied. It is not clear from the methods what the scale of the neighbourhood analysis used to calculate slope (e.g. 3x3 moving window?), so perhaps this point could be dealt with by providing further details in that section? This also applies for the field measurements form the transects. It is not clear how slope was calculated along the 12-32.5m long transects (was the entire length of the transect used?) nor how this scale related to the scale at which forest management prescriptions are intended to be applied in a forest block.

2. If length is an issue for the editor, the section titled "Comparisons with analysis of steep slopes by VicForests" could be edited back as it provides a lot of detail about the authors communications with the agency which is not essential to the paper's important findings.

Reviewer #2: This study addresses an important aspect of environmental management that is open to misuse, clarifying critical details with sound data. It’s important that it is published, but I think it could be improved with some minor changes.

The main issue is that the framing of the study needs to be clearer. Given that DEMs have been use for decades for similar purposes, it’s unclear why this is an important question until the reader reaches the Discussion. The core of it is in line 564, where the OCR claimed that these DEMs overstated the slope. I think that needs to be up-front in the Intro – concerns have been expressed that DEMs overstate the slope.

A second although not essential issue is that the paper would be stronger if it investigated a mechanism, so that the findings are more widely applicable. I can see two possible options from the findings:

1. Lower resolution DEMs (larger cells) smooth the landscape by averaging the contents of the cell. Where there is greater change in slope within a cell (eg gullies), you could expect the DEM to underestimate the slope (eg the 30m DEM).

2. The accuracy of point heights used in a DEM can cause errors in slope calculations, but unless such errors have a systematic trend, this should just create noise rather than bias (eg the 10m DEM)

You could either test these as hypotheses or summarise them as key findings.

Other minor issues:

60: Delete ‘locating’

82: Saying that DEMs are already considered essential seems to conflict with the idea that you’re presenting something novel. That needs to be said in context – “The use of DEMs is essential for forest management planning to identify steep slopes and exclude logging from these areas, however, some have contended that…”

88: Add comma between ‘LiDAR and’

109: Unless this is the primary aim of the paper, maybe state it after the other aim(s). The first of these should be to investigate whether DEMs distort slopes as claimed.

152: Practice, not practices

202-205: Seems to be repeating earlier parts of the paragraph a bit, maybe simplify the paragraph as a whole.

212-214: A minimum patch size should be clearly specified that breaches the code. If there is no published value, make this clear and say that you’re using 5m because there isn’t a standard. That could also feed into the Discussion, highlighting that systems are open to abuse if no patch size has been specified.

272: Drop the final ‘the’

367: ‘Consisted of’ implies that it was all 1m patches. Perhaps change to ‘included’.

468: Lose the period after the first ‘the’

568-570: This is the core finding and is important. I’d open the results with this.

630ff: It’s unclear how this section fits the paper overall. While they seem like questionable actions, I’d only include them if you can show clearly how they breach your findings.

683: Such ‘data’

701: “compliance with laws preventing logging on steep slopes is require[d]”

6. PLOS authors have the option to publish the peer review history of their article (what does this mean?). If published, this will include your full peer review and any attached files.

Reviewer #1: No

Reviewer #2: No

---

## [Author Response · Author response to Decision Letter 0]

7 Apr 2022

Professor David Lindenmayer

Dr Chris Taylor

Fenner School of Environment and Society

The Australian National University

Canberra, ACT, 2601

4 March 2022

Re: Revisions to PONE-D-21-34995 - The use of spatial data and satellite information in legal compliance and planning in forest management

Dear Professor RunGuo Zang,

We enclose our substantially revised paper PONE-D-21-34995 - The use of spatial data and satellite information in legal compliance and planning in forest management for re-consideration for publication in PLOS ONE. 

We have used the comments of the two referees to help guide our revisions – and these changes have helped further strengthen the paper. In the remainder of this letter we provide a point-by-point response to each of the comments of both reviewers. 

We trust that this letter and our revised manuscript will be received favourably and look forward to hearing from you in the near future. 

Yours sincerely,

David Lindenmayer

(on behalf of both authors)

 

COMMENTS FROM REVIEWERS

Comments from Reviewer #1:

This is an interesting paper which presents a very detailed applied study concerning the application of digital terrain analysis, specifically topographic slope angle, to forest management prescriptions. The results are novel in terms of the context, namely, the efficacy of differently grained DEMs and the slope derivative to inform the monitoring and evaluation of forest management prescriptions designed to protect water catchments. The paper presents an extremely thorough empirical analysis as well as providing an in-depth discussion of the practical relevance of the results. The paper warrants publishing in PLOS ONE subject to a minor revision to take account of the following points. 

Response from authors

We thank Reviewer #1 for their very supportive comments. We do recognize that Reviewer #1 has made some insightful comments and we have worked hard to address their suggestions to help improve the manuscript. 

Comments from Reviewer #1:

L60 - “isn’t” should be “is not” 

Response from authors

We have corrected this typographical error. 

Comments from Reviewer #1:

L62 – remove brackets as this is a key point, not an aside 

Response from authors

We have removed the brackets as suggested. 

Comments from Reviewer #1:

L72 does “best” = “most cost effective”? What are criteria for assessing what is “best” here? Response from authors

We have replaced the term “best” with most cost-effective as suggested by Referee #1. 

Comments from Reviewer #1:

L74 – strictly speaking a DEM is not remote sensing data. It is a spatial data layer modelled from a satellite-based or otherwise air-borne active sensor. 

Response from authors

This was a fair point made by Reviewer #1 and we amended the text to reflector this fact. 

Comments from Reviewer #1:

L77-78 – this sentence needs to be rewritten to make clearer the meaning given it is the basis for the relevance of the paper’s analyses. 

Response from authors

We have reworked the sentence as requested by Reviewer #1. 

Comments from Reviewer #1:

L 105-108, 109-112 – these read more like conclusions, rather than introductory statements, as they are anticipating the results. 

Response from authors

As suggested by Reviewer #1, we have now removed the text on Lines 109-113 as they did indeed anticipate the results. 

Comments from Reviewer #1:

L 192 – how was slope calculated in Arc GIS? I.e. different functions can be used to calculate gradients in the X and Y direction. 

Response from authors

The ArcGIS Slope tool accounts for slope gradients in both the x and y directions. We have now made this clear in our text.

Comments from Reviewer #1:

L221 – can the authors clarify that there were Lidar shots for every grid cell? If not, then some interpolation would have been required. 

Response from authors

We generated LiDAR DEM from the LAS files to a 1 metre grid raster in ArcGIS. The point separation ranged between 10 cm to 25 cm. Where cells did not have points, we triangulated across void areas and used linear interpolation on the triangulated value to determine the cell value (ESRI 2021). We have now added this to our text. 

Reference

ESRI (2021), LAS Dataset To Raster, https://desktop.arcgis.com/en/arcmap/latest/tools/conversion-toolbox/las-dataset-to-raster.htm, accessed 24 February 2022

Comments from Reviewer #1:

L224 – was there a rationale behind the class intervals? 

Response from authors

The class intervals used in our analysis were based on the Absolute Risk Rating Methodology previously used in the Forest Audit Program for the Victorian Government (URS Australia 2011). We have now added this this to our text. 

Reference

URS Australia (2011), Report Environmental Audit - Forest Audit Program Module 6 - Harvesting Performance, URS Australia Pty Limited. 

Comments from Reviewer #1:

L268 – it is not clear why you would need to test for spatial autocorrelations given all the data being analysed are DEMs on a regular grid, it is not clear why autocorrelation would be an issue?? What was the autocorrelation of concern? 

Response from authors

We have removed this section of the paper. Upon review, there were some issues with the spatial autocorrelation because we are analysing individual cutblocks and these are already clustered around access (i.e. roads) and specific forest types.

Comments from Reviewer #1:

L346 – I think the differences between the values of the differently scaled slope grides are better described as differences and not errors. I say this because slope can be accurately calculated at a range of scales for the same location and differ in their values. 

Response from authors

We have removed the term “errors” from our revised manuscript. We have replaced the term with ‘difference’ where relevant. The term ‘error’ is now retained only for our descriptions of guidance provided by the data providers for the datasets themselves, i.e. “For Australia, the horizontal positional error for the SRTM DEM is within 7.2 m and the elevation error is within 9.8 m”. These positional errors were based on in field measurements used in the DEM guidance documents that were used for comparison with the DEMs themselves. 

Comments from Reviewer #1:

L360 - clarify that this result was found from all three scaled slope grids, or if not, which scale was used. 

Response from authors

We have now added the following amended text: “The slope raster we generated from LiDAR DEM showing all slopes >30° (including small areas at 1.0m2) indicated that all cut blocks in the Thomson water supply protection area contained slopes >30° and all but one cut block supported slopes >30° in the Upper Goulburn water supply protection area (Table 1).” 

Comments from Reviewer #1:

General comments: 

1. It would be helpful if the authors could clarify if all three scales are relevant to the scale at which the forest management guidelines are intended to be applied. For example, I don’t think that forest management prescriptions are meant to be applied at a 1m resolution. Wouldn’t the appropriate scale of analysis be the slope calculated at a scale that captures the variation within a logged coup; which I assume is equivalent to what the authors call a cut block? In any case, some discussion is warranted of the scale at which slope is calculated compared with the scale at which forest management prescriptions are applied. It is not clear from the methods what the scale of the neighbourhood analysis used to calculate slope (e.g. 3x3 moving window?), so perhaps this point could be dealt with by providing further details in that section? This also applies for the field measurements form the transects. It is not clear how slope was calculated along the 12-32.5m long transects (was the entire length of the transect used?) nor how this scale related to the scale at which forest management prescriptions are intended to be applied in a forest block. 

Response from authors

This is an important point raised by Reviewer #1. The 2014 Management Standards and Procedures did not specify a minimum area under Clause 3.5 addressing the logging of steep slopes that would constitute a breach. The Office of the Conservation Regulator later provided a description of an average slope as the average slope of the topographic feature of interest that is subject to assessment (Zanini 2020). Again, this did not specify a minimum distance. In the absence of such specification, we used an average moving circle of a 5m radius to calculate an average slope. This used the local neighbourhood tool in ArcGIS called focal statistics. This provided for an average distance of at least 10m in distance on where a steep slope >30° could be identified. This was applied only to the slope raster generated from the LiDAR DEM, because it consisted of a cell size of 1m. This was not applied to the slope raster generated from the VicMap Elevation DTM and the SRTM DEM, because these had a respective cell size of 10m and ~30m, respectively. We contended that these datasets and associated respective cell sizes provide a good indication of where a slope exceeded the limits specified under the Management Standards and Procedures. 

Comments from Reviewer #1:

2. If length is an issue for the editor, the section titled "Comparisons with analysis of steep slopes by VicForests" could be edited back as it provides a lot of detail about the authors communications with the agency which is not essential to the paper's important findings.

Response from authors

This was a fair suggestion from Reviewer #1 and we have reduced the length of this section as suggested. 

 

COMMENTS FROM REVIEWER #2

Comments from Reviewer #2:

Reviewer #2: This study addresses an important aspect of environmental management that is open to misuse, clarifying critical details with sound data. It’s important that it is published, but I think it could be improved with some minor changes. 

Response from authors

We thank Reviewer #2 for their support for our paper. We have used the suggestions of Reviewer #2 to help further strengthen the paper; we are grateful for the insights they have provided. 

Comments from Reviewer #2:

The main issue is that the framing of the study needs to be clearer. Given that DEMs have been use for decades for similar purposes, it’s unclear why this is an important question until the reader reaches the Discussion. The core of it is in line 564, where the OCR claimed that these DEMs overstated the slope. I think that needs to be up-front in the Intro – concerns have been expressed that DEMs overstate the slope. 

Response from authors

This was an astute point made by Reviewer #2 and we have now revised part of the Introduction to make it clearer that testing slopes with DEMs is important given claims by the regulator that slopes are overstated. 

Comments from Reviewer #2:

A second although not essential issue is that the paper would be stronger if it investigated a mechanism, so that the findings are more widely applicable. I can see two possible options from the findings: 

1. Lower resolution DEMs (larger cells) smooth the landscape by averaging the contents of the cell. Where there is greater change in slope within a cell (eg gullies), you could expect the DEM to underestimate the slope (eg the 30m DEM).

2. The accuracy of point heights used in a DEM can cause errors in slope calculations, but unless such errors have a systematic trend, this should just create noise rather than bias (eg the 10m DEM). 

You could either test these as hypotheses or summarise them as key findings. 

Response from authors

These are important points. We have added these comments to our discussions.

Comments from Reviewer #2:

Other minor issues: 

60: Delete ‘locating’

Response from authors

We have delete the word “locating” as suggested by Reviewer #2. 

Comments from Reviewer #2:

82: Saying that DEMs are already considered essential seems to conflict with the idea that you’re presenting something novel. That needs to be said in context – “The use of DEMs is essential for forest management planning to identify steep slopes and exclude logging from these areas, however, some have contended that…”

Response from authors

This recommended rewording from Reviewer #2 was very helpful and we have now revised the text to better frame the Introduction. 

Comments from Reviewer #2:

88: Add comma between ‘LiDAR and’

Response from authors

We have corrected this typographical error as identified by Reviewer #2. 

Comments from Reviewer #2:

109: Unless this is the primary aim of the paper, maybe state it after the other aim(s). The first of these should be to investigate whether DEMs distort slopes as claimed. 

Response from authors

We agree with Reviewer #2 and removed the final paragraph from the reframed Introduction section. 

Comments from Reviewer #2:

152: Practice, not practices 

Response from authors

We have corrected this typographical error. 

Comments from Reviewer #2:

202-205: Seems to be repeating earlier parts of the paragraph a bit, maybe simplify the paragraph as a whole. 

Response from authors

We carefully examined the length of this paragraph as recommended by Reviewer #2. However, shortening it would have deleted key details on the spatial data approach that we employed and would have become confusing to readers. 

Comments from Reviewer #2:

212-214: A minimum patch size should be clearly specified that breaches the code. If there is no published value, make this clear and say that you’re using 5m because there isn’t a standard. That could also feed into the Discussion, highlighting that systems are open to abuse if no patch size has been specified. 

Response from authors

This issue was also identified by Reviewer #1. We have added additional text in our Methods section clarifying that the 2014 Management Standards and Procedures did not specify a minimum area threshold for steep slope measurements and the subsequent guidance provided by the OCR, while providing further clarification, also did not specify a minimum threshold. We used a 5m radius for a moving average across our analysis area would identify areas upon where an average slope was >30°. This is based on our own contentions. 

Comments from Reviewer #2:

272: Drop the final ‘the’ 

Response from authors

We have amended the text as suggested by Reviewer #2. 

Comments from Reviewer #2:

367: ‘Consisted of’ implies that it was all 1m patches. Perhaps change to ‘included’. 

Response from authors

We have changed the text to use the word “included” as suggested by Reviewer #2. 

Comments from Reviewer #2:

468: Lose the period after the first ‘the’ 

Response from authors

We have corrected the text. 

Comments from Reviewer #2:

568-570: This is the core finding and is important. I’d open the results with this. 

Response from authors

This was a good suggestion from Reviewer #2 and we have now re-ordered the sequence of key findings in the Results section to include this core finding. 

Comments from Reviewer #2:

630ff: It’s unclear how this section fits the paper overall. While they seem like questionable actions, I’d only include them if you can show clearly how they breach your findings. 

Response from authors

This section is important because it highlights that statements by Victorian Government agencies that only limited areas above 30 degrees in slope have been logged are inconsistent with the spatial data that we have presented. However, we recognize that both referees considered this section was too long and we have now reduced the length of the text in this section considerably. 

Comments from Reviewer #2:

683: Such ‘data’ 

Response from authors

We have corrected the text. 

Comments from Reviewer #2:

701: “compliance with laws preventing logging on steep slopes is require[d]”

Response from authors

We have corrected this typographical error. 

---

## [Editor Report · Decision Letter 1]

20 Apr 2022

The use of spatial data and satellite information in legal compliance and planning in forest management

PONE-D-21-34995R1

Dear Dr. Lindenmayer,

We’re pleased to inform you that your manuscript has been judged scientifically suitable for publication and will be formally accepted for publication once it meets all outstanding technical requirements.

Kind regards,

RunGuo Zang

Academic Editor

PLOS ONE

Additional Editor Comments (optional):

accept